

**Concomitant ocean acidification and increasing total alkalinity at a coastal site in the NW**
**Mediterranean Sea (2007-2015)**
Lydia Kapsenberg[1], Samir Alliouane[1], Frédéric Gazeau[1], Laure Mousseau[1], and Jean-Pierre
Gattuso[1,2,§]
[1]Sorbonne Universités, Université Pierre et Marie Curie-Paris 6, CNRS-INSU, Laboratoire
d'Océanographie de Villefranche, 06230, Villefranche-sur-Mer, France
[2]Institute for Sustainable Development and International Relations, Sciences Po, 27 rue Saint
Guillaume, F-75007 Paris, France
[§]Corresponding author
E-mail: gattuso@obs-vlfr.fr
Phone: +33 4 93 76 38 59



**Abstract.** Monitoring global ocean change is necessary in coastal zones due to their physical and
biological complexity. Here, we document changes in coastal carbonate chemistry at the time-
series station, Point B, in the NW Mediterranean Sea, from 2007 through 2015, at 1 and 50 m.
The rate of surface ocean acidification (-0.0028 ± 0.0003 units $pH_T$ $yr^{-1}$) was faster-than-
expected based on atmospheric carbon dioxide forcing alone. Changes in carbonate chemistry
were predominantly driven by an increase in total dissolved inorganic carbon ($C_T$, +2.97 ± 0.20
$\mu$mol $kg^{-1}$ $yr^{-1}$), > 50 % of which was buffered by a synchronous increase in total alkalinity ($A_T$,
+2.08 ± 0.19 $\mu$mol $kg^{-1}$ $yr^{-1}$). The increase in $A_T$ was unrelated to salinity and its cause remains to
be identified. Interestingly, concurrent increases in $A_T$ and $C_T$ were most rapid from May to July.
Changes at 50 m were slower compared to 1 m. It seems therefore likely that changes in coastal
$A_T$ cycling via a shallow coastal process gave rise to these observations. This study exemplifies
the importance of understanding coastal ocean acidification through localized biogeochemical
cycling that extends beyond simple air-sea gas exchange dynamics, in order to make relevant
predictions about future coastal ocean change and ecosystem function.
**Keywords** – global ocean change, ocean acidification, time-series, pH, alkalinity, dissolved
inorganic carbon, $pCO_2$, Mediterranean Sea, near-shore



## 1. Introduction

Maintaining time-series of oceanographic data is essential for understanding anthropogenic change in the ocean (Tanhua et al., 2013). On land, fossil fuel burning, cement production and land use changes have contributed ~580 Gt carbon to the atmosphere during the period 1750-2013 (Le Quéré et al., 2015). An estimated 29 % of this anthropogenic carbon is absorbed by the ocean in the form of carbon dioxide ($CO_2$; Le Quéré et al., 2015), and causing global changes to the ocean carbonate system. Absorption of $CO_2$ by seawater produces carbonic acid, which decreases seawater pH, and is of great concern for biological processes and marine ecosystems (Doney et al., 2009; Gattuso and Hansson, 2011; Pörtner et al., 2014). Since the preindustrial era, global mean ocean pH has declined by 0.1 (Rhein et al., 2013). Due to the declining trend of ocean pH with increasing anthropogenic $CO_2$, the process is termed 'ocean acidification', but this expression represents a suite of chemical changes, including increases in total dissolved inorganic carbon ($C_T$) and partial pressure of $CO_2$ (pCO$_2$) and decrease in calcium carbonate saturation states ($\Omega$, aragonite and calcite; Dickson, 2010). Rates of ocean acidification differ by region and range from -0.0013 units pH yr$^{-1}$ (South Pacific) to -0.0026 units pH yr$^{-1}$ (Irminger Sea, North Atlantic) and are reviewed in Bates et al. (2014). Such time-series remain spatially limited, especially in coastal regions which provide valuable ecosystem services (Barbier et al., 2011; Costanza et al., 1997) and are under high anthropogenic impact (Halpern et al., 2008). Here, we present the first ocean acidification time-series at weekly frequency for a coastal site in the Mediterranean Sea.

Compared to the global ocean, marginal seas serve a critical role in anthropogenic $CO_2$ storage via enhanced $CO_2$ uptake and export to the ocean interior (Lee et al., 2011). As a marginal sea, the Mediterranean Sea has a naturally high capacity to absorb but also buffer





anthropogenic $CO_2$ (Álvarez et al., 2014; Palmiéri et al., 2015). This is primarily due to the high
total alkalinity ($A_T$) of Mediterranean waters and overturning circulation (Lee et al., 2011;
Palmiéri et al., 2015; Schneider et al., 2010). In the Mediterranean Sea, the salinity-$A_T$
relationship is driven by the addition of river discharge and Black Sea input, which are generally
both high in $A_T$ (Copin-Montégut, 1993; Schneider et al., 2007). Combined with evaporation,
this results in higher $A_T$ and salinity in the Mediterranean Sea compared to the Atlantic
Mediterranean source water (Jiang et al., 2014). On average, Mediterranean Sea $A_T$ is 10 %
higher than in the global ocean (Palmiéri et al., 2015). The surface ocean acidification rate,
estimated at $\Delta pH_T$ of -0.08 since 1800, is comparable to that of the global ocean despite a 10%
greater anthropogenic carbon inventory (Palmiéri et al., 2015). Due to its important role in
carbon sequestration and ecological sensitivity to global ocean change with economic impacts
(Lacoue-Labarthe et al., 2016), the Mediterranean Sea is a key location for time-series
measurements.

Over the last few years, numerous studies have estimated ocean acidification rates across

the Mediterranean Sea (Table 1). Together, these studies cover various study periods with a
range of techniques yielding different results. For example, estimates of change in pH of bottom
waters since the preindustrial era range between -0.005 to -0.06 (Palmiéri et al., 2015) and as
much as -0.14 (Touratier and Goyet, 2011). Techniques for estimating ocean acidification in the
Mediterranean Sea thus far include: (1) hind-casting, using high-resolution regional circulation
models (Palmiéri et al., 2015), the TrOCA approach as applied to cruise-based profile data
(Krasakopoulou et al., 2011; Touratier and Goyet, 2011; Touratier et al., 2016) and others
(Howes et al., 2015), (2) partially reconstructed time-series (Marcellin Yao et al., 2016), (3)
comparative study periods (Luchetta et al., 2010; Meier et al., 2014), and (4) sensor-based





observations over a short study period (Flecha et al., 2015). Ocean acidification time-series of
consistent sampling over many years are lacking for the Mediterranean Sea (The MerMex
Group, 2011), especially along the coast where river discharge influences the carbonate system
(Ingrosso et al., 2016).

Compared to the open ocean, shallow coastal sites exhibit natural variability in carbonate

chemistry over annual timeframes (Hofmann et al., 2011; Kapsenberg and Hofmann, 2016;
Kapsenberg et al., 2015), complicating the detection and relevance of open ocean acidification in
isolation of other processes (Duarte et al., 2013). Variability stems from both physical (e.g.,
upwelling, river discharge; Feely et al., 2008; Vargas et al., 2016) and biological processes (e.g.,
primary production, respiration, net calcification). Within watersheds, coastal carbonate
chemistry is affected by eutrophication (Borges and Gypens, 2010; Cai et al., 2011),
groundwater supply (Cai et al., 2003), and land use and rain influence on river alkalinity
(Raymond and Cole, 2003; Stets et al., 2014). Over longer periods, pH can also be influenced by
atmospheric deposition (Omstedt et al., 2015). Introduction of nutrient-rich upwelled- or fresh-
water masses influences biological processes and carbonate chemistry at higher frequencies.
Through primary production and respiration, coastal ecosystems produce pH fluctuations over
hours (e.g., seagrass, kelp) to months (e.g., phytoplankton blooms; Kapsenberg and Hofmann,
2016). Due to existing pH variability in coastal seas, it is necessary to quantify high-frequency
pH variability in order to interpret the pH changes inferred from lower-frequency sampling, at
time-series stations.

In this study, we present the first complete time-series data quantifying the present-day

ocean acidification rate for a coastal site in the Mediterranean Sea, based on weekly
measurements of $A_T$ and $C_T$ sampled from 2007 through 2015. For a subset of this time-series,



we documented pH variability using a SeaFET™ Ocean pH Sensor in order to assess hourly pH
variability. For comparison and consistency with other ocean acidification time-series around the
world, we report rates of change based on anomalies (Bates et al., 2014).

**2. Materials and methods**
**2.1. Site description**

A carbonate chemistry time-series was initiated in 2007 in the NW Mediterranean Sea at

the entrance of the Bay of Villefranche-sur-Mer, France (Fig. 1): Point B station (43.686° N,
7.316° E, 85 m bottom depth). A second site, Environment Observable Littoral buoy (EOL,
43.682° N, 7.319° E, 80 m bottom depth), was used for pH sensor deployment starting in 2014.
These two sites are 435 m apart. The site Point B is an historical sampling point, since 1957,
regarding several oceanographic parameters. A full site description and research history has been
detailed by De Carlo et al. (2013). Briefly, the Bay is a narrow north-south facing inlet with steep
bathymetry and estimated volume of 310 million m$^3$. The surrounding region is predominately
composed of limestone with a series of shallow, submarine groundwater karst springs (Gilli,
1995). The North current, a major and structuring counter-clockwise current in the Ligurian Sea,
can sometimes flow close to Point B. The Bay can also be, on occasion, influenced by local
countercurrents. Both of these hydrodynamics movements have signatures of riverine discharge,
which for the Mediterranean Sea are generally high in $A_T$ (Copin-Montégut, 1993; Schneider et
al., 2007). For example, the Paillon River, 4 km west of Point B and whose plume on occasion
reaches into the Bay (L. Mousseau, pers. obs.), was sampled on 18 Aug 2014 and had a $A_T$ of
1585 ± 0.1 µmol kg$^{-1}$ ($N$=2, J.-P. Gattuso, unpubl.). Due to low primary productivity, seasonal



warming drives the main annual variability in carbonate chemistry at this location (De Carlo et
al., 2013).

**2.2. Point B data collection, processing, and analysis**

To document long-term changes in ocean carbonate chemistry at Point B, seawater was

sampled weekly starting in January 2007. Samples were collected at 1 and 50 m, using a 12-L
Niskin bottle at 9:00 local time. Seawater was transferred from the Niskin bottle to 500 mL
borosilicate glass bottles and fixed within an hour via addition of saturated mercuric chloride for
preservation of carbonate parameters, following recommendations by Dickson et al. (2007).
Duplicate samples were collected for each depth. For each sampling event, CTD casts were
performed either with a Seabird 25 or Seabird 25+ profiler whose sensors are calibrated at least
every two years. Accuracy of conductivity (SBE4 sensor) and temperature (SBE3 sensor)
measurements from CTD casts were 0.0003 S m$^{-1}$ and 0.001°C, respectively.

Within 6 months of collection, bottle samples were analyzed for $C_T$ and $A_T$ via

potentiometric titration following methods described by Edmond (1970) and DOE (1994), by
*Service National d'Analyse des Paramètres Océaniques du CO$_2$*, at the Université Pierre et Marie
Curie in Paris, France. Precision of $C_T$ and $A_T$ was less than 3 µmol kg$^{-1}$, and the average
accuracy was 2.6 and 3 µmol kg$^{-1}$, as compared with seawater certified reference material
(CRM) provided by A. Dickson (Scripps Institution of Oceanography). Only obvious outliers
were omitted from the analyses: three $C_T$ values at 1 m (> 2300 µmol kg$^{-1}$), one $A_T$ value at 1 m
(> 2900 µmol kg$^{-1}$), one $A_T$ value at 50 m (< 2500 µmol kg-1). The $C_T$ and $A_T$ measurements on
replicates bottle samples were averaged for analyses.



Calculations of the carbonate system parameters were performed using the R package
seacarb version 3.1 with $C_T$, $A_T$, temperature and salinity as inputs (Gattuso et al., 2016). Total
concentrations of silicate ($SiOH_4$) and phosphate ($PO_4^{3-}$) were used when available from Point B
(L. Mousseau, unpubl.). Detection limits for nutrients were 0.05 µM for $SiOH_4$ and 0.003 to
0.006 µM for $PO_4^{3-}$; relative precision of these analyses is 5-10 % (Aminot and Kérouel, 2007).
Total boron concentration was calculated from salinity using the global ratio determined by Lee
et al. (2010). The following constants were used: $K_1$ and $K_2$ from Lueker et al. (2000), $K_f$ from
Perez and Fraga (1987), and $K_s$ from Dickson (1990). Reported measured parameters are
temperature, salinity, $A_T$, and $C_T$, and derived parameters are $pH_T$ (total hydrogen ion scale), $pH_T$
normalized to 25 ºC ($pH_{T25}$), $pCO_2$, and aragonite ($\Omega_a$) and calcite ($\Omega_c$) saturation states. Salinity-
normalized changes in $A_T$ ($nA_T$) and $C_T$ ($nC_T$) were calculated by dividing by salinity and
multiplying by 38. Except for $pH_{T25}$, all parameters are reported at *in situ* temperatures.
The average uncertainties of the derived carbonate parameters were calculated according
to the Gaussian method (Dickson and Riley, 1978) implemented in the "errors" function of the R
package seacarb 3.1 (Gattuso et al., 2016). The uncertainties are ±2.7 x $10^{-10}$ mol $H^+$ (about
0.015 units $pH_T$), ±15 µatm $pCO_2$, and ±0.1 unit of the aragonite and calcite saturation states.
To quantify interannual changes in carbonate parameters, the data were detrended for
seasonality by subtracting monthly means from the time-series following methods in Bates et al.
(2014). The resulting anomalies were analyzed using a linear regression. All analyses were
performed in R (R Core Team, 2016).

**2.3. Deconvolution of $pH_T$ and $pCO_2$**



To identify proportional contributions of various drivers to ocean acidification trends at
Point B, deconvolution of time-series $pH_T$ and $pCO_2$ was performed following methods from
García-Ibáñez et al. (2016) for observations at 1 m. The equation is described below for $pH_T$,
where changes in $pH_T$ are driven by changes in temperature ($T$), salinity ($S$), $A_T$, and $C_T$, over
time ($t$), according to the following model:
$$\frac{dpH_T}{dt} = \frac{\partial pH_T}{\partial T}\frac{dT}{dt} + \frac{\partial pH_T}{\partial S}\frac{dS}{dt} + \frac{\partial pH_T}{\partial A_T}\frac{dA_T}{dt} + \frac{\partial pH_T}{\partial C_T}\frac{dC_T}{dt} \qquad (1)$$
Here, $\frac{\partial pH_T}{\partial var}\frac{dvar}{dt}$ represents the slope contribution of changing $var$ to the estimated change
in $pH_T$, where $var$ is either $T$, $S$, $A_T$, or $C_T$. The rate of pH change due to $var$ ($\frac{\partial pH_T}{\partial var}\frac{dvar}{dt}$) was
estimated by calculating $pH_T$ using the true observations of $var$ but monthly mean values of the
other three variables, and regressing it to time. The calculation was repeated for $pCO_2$ ($\frac{dpCO_2}{dt}$) in
order to compare the rate of increase with that of atmospheric $CO_2$.
As a sub-component of $\frac{\partial pCO_2}{\partial C_T}\frac{dC_T}{dt}$ , the rate of anthropogenic $CO_2$ increase was estimated
from atmospheric $CO_2$ concentrations nearest to Point B (Plateau Rosa, Italy, courtesy of the
World Data Center for Greenhouse Gases, http://ds.data.jma.go.jp/gmd/wdcgg/). For these data,
missing daily values were linearly interpolated. A linear regression was performed where the
slope represents the rate of $CO_2$ increase in the atmosphere. Finally, to help identify different
processes that might have contributed to the observed trends, linear regressions were performed
on change in $A_T$ and $C_T$ by month and on the salinity-$A_T$ relationship by year.

**2.4. SeaFET data collection, processing, and analysis**
To capture pH variability at higher-than-weekly sampling frequencies, a SeaFET™
Ocean pH sensor (Satlantic) was deployed on the EOL buoy (435 m from the Point B sampling



site) starting in June 2014, at 2 m depth. Autonomous sampling was hourly and deployment
periods ranged between 1 and 3 months. Field calibration samples for pH were collected weekly,
using a Niskin bottle next to SeaFET within 15 min of measurement. This sampling scheme was
sufficient for this site as there is no large high-frequency pH variability. Unlike Point B
sampling, SeaFET calibration samples were processed for pH using the spectrophotometric
method (Dickson et al., 2007) with purified m-cresol purple (purchased from the Byrne lab,
University of South Florida). *In situ* temperature, salinity, and $A_T$ measured at Point B, within 30
min of the SeaFET sampling, were used to calculate *in situ* $pH_T$ of the calibration samples.
SeaFET voltage was converted to $pH_T$ using the respective calibration samples for each
deployment period, following the methods and code described in Bresnahan et al. (2014) but
adapted for use in R.

The estimated standard uncertainty in SeaFET $pH_T$ is ±0.01 and was calculated as the

square root of the sum of each error squared. The sources of errors are: measurement error of
spectrophotometric pH (±0.004, *N*=68 mean SD of 5 replicate measurements per calibration
sample for samples collected between 16 July 2014 and 3 May 2016), spatio-temporal mismatch
sampling at EOL (±0.007, mean offset of $pH_T$ of the calibration samples from calibrated time-
series), and variability in purified m-cresol dye batch accuracy as compared to Tris buffer CRM
pH (±0.006, mean offset of $pH_T$ of the spectrophotometric measurement of Tris buffer from the
CRM value).

**3. Results**
**3.1. Time-series trends**



At Point B from January 2007 to December 2015, >400 samples were collected for
carbonate chemistry at both 1 and 50 m. Anomaly trends detected at 1 m (Fig. 2, Table 2) were
also significant at 50 m (Table S1, Fig. S1), with the exception that salinity increased at 50 m
($0.0063 \pm 0.0020$ units $yr^{-1}$, $P = 0.002$). Changes in carbonate chemistry were faster at 1 m
compared to 50 m, with the exception of salinity and temperature. The warming rate at 50 m was
22 % greater compared to 1 m, mostly due to increasing summer temperatures since 2007.
Analyses for 50 m are available (Table S1, Fig. S1) but here, we focus on results from 1 m unless
explicitly specified otherwise (Fig. 2, Table 2). For time-series anomalies at 1 m, carbonate
chemistry anomalies were significant for $pH_T$ ($-0.0028 \pm 0.0003$ units $yr^{-1}$, $N$=412), $A_T$ ($+2.08 \pm$
$0.19$ $\mu$mol $kg^{-1}$ $yr^{-1}$, $N$=417), $C_T$ ($+2.97 \pm 0.20$ $\mu$mol $kg^{-1}$ $yr^{-1}$, $N$=416), $pCO_2$ ($+3.53 \pm 0.39$ $\mu$atm
$yr^{-1}$, $N$=412), and $\Omega_a$ ($-0.0064 \pm 0.0015$ units $yr^{-1}$, $N$=412). At the same time, temperature
anomaly increased ($+0.072 \pm 0.022$ ºC $yr^{-1}$, $N=413$), but this significance was lost with the
exclusion of the year 2015 (analysis not shown; changes in other carbonate chemistry parameters
remained significant). No significant change in the salinity anomaly was detected ($P$=0.702,
$N$=417).
Strong seasonal cycles in carbonate parameters were present at Point B at 1 m (Fig. 3).
Calculated monthly means (2007-2015) are described briefly and listed in Table S2. Mean
temperature range was 11.2 ºC with a maximum at $24.77 \pm 1.35$ ºC in August and minimum of
$13.58 \pm 0.41$ ºC in February. The range in $A_T$ was $+19$ $\mu$mol $kg^{-1}$ from June to September. The
$C_T$ range was 33 $\mu$mol $kg^{-1}$ with a peak in late winter and minimum values in August and
October. The opposing seasonal cycles of temperature and $C_T$, due to summer warming
coinciding with the period of peak primary productivity (De Carlo et al. 2013), resulted in an
annual $pH_T$ range of 0.12 with peak pH in late winter ($8.14 \pm 0.01$, February and March) and





minimum pH in summer (8.02 ± 0.03, July and August). The corresponding $pCO_2$ range was
+128 µatm from February to August. The monthly means were subtracted from the time-series to
calculate anomalies.

**3.2. Deconvolution of $pH_T$ and $pCO_2$**

Deconvolutions of pH and $pCO_2$ are presented in Table 3. For both $\frac{dpH_T}{dt}$ and $\frac{dpCO_2}{dt}$,

temperature and salinity had contributing (approx. 35% and 5% respectively) but non-significant
effects. The 2007-2015 observation period is likely not long enough to detect significance of the
relatively large temperature contribution to $\frac{dpH_T}{dt}$. The predominant driver of $\frac{dpH_T}{dt}$ and $\frac{dpCO_2}{dt}$ was
the increase in $C_T$ ($P \ll 0.001$), 56 and 60 % of which was compensated for by a significant
increase in $A_T$ ($P = 0.002$), respectively.

Atmospheric $CO_2$ at Plateau Rosa increased by 2.02 ± 0.03 ppm $yr^{-1}$ ($F_{1,3285} = 5852.43$, $P$

$\ll 0.001$, $R^2$ 0.64), with an anomaly rate of 2.08 ± 0.01 ppm $yr^{-1}$ ($F_{1,3285} = 46649.79$, $P \ll 0.001$,
$R^2$ 0.93) during the study period 2007-2015, and represents the anthropogenic $CO_2$ forcing of
ocean acidification. Considering the error associated with deconvolution of $pCO_2$ and assuming
air-sea $CO_2$ equilibrium, atmospheric $CO_2$ increase represents 31 to 44 % of the total $C_T$
contribution ($\frac{\partial pCO_2}{\partial C_T} \frac{dC_T}{dt}$) to $\frac{dpCO_2}{dt}$. This leaves 56 to 69 % of the total $C_T$ contribution to $pCO_2$
unaccounted for. As $A_T$ is not influenced by addition of anthropogenic $CO_2$ to seawater, the next
question was, thus, whether or not the changes in $A_T$ and $C_T$ were process-linked. Regressions of
annual monthly observations of $A_T$ and $C_T$ revealed similar seasonal cycles for both parameters
(Fig. 4). The fastest increases occurred simultaneously for both parameters, which peaked in
June. The smallest changes occurred in January. Significant increases in $A_T$ occurred from May



to July. Significant increases in $C_T$ occurred from February-August and in November. Results of
regression analyses on monthly changes are listed in Table S3.

**3.3. Salinity and $A_T$ relationships**

Given the coastal locale and the increase in $A_T$, it is interesting to look at the interannual
variability of the relationship between salinity and $A_T$ (Fig. 5). With the exception of 2007,
salinity was a poor predictor of $A_T$. The $R^2$ value for each annual salinity-$A_T$ linear regression
ranged from 0.00 (in 2013) to 0.87 (in 2007) with y-intercepts ($A_{T0}$, the freshwater end-member
alkalinity) ranging between -176 µmol kg$^{-1}$ (in 2007) and 2586 µmol kg$^{-1}$ (in 2013). The
interannual variability of the salinity-$A_T$ relationship was driven by the variability in $A_T$ observed
at salinity < 38.0 that was present from November through July.
The salinity cycle (monthly means) at Point B was small and ranged from 37.64 ± 0.26 to
38.21 ± 0.11 from May to September, following freshwater input in winter and spring and
evaporation throughout summer and fall (Fig. 3). Highest (> 38.0) and most stable salinity
observations were made in August through October, which coincided with the period of
maximum $A_T$ (2562 and 2561 ± 9 µmol kg$^{-1}$ in September and October, respectively). Minimum
$A_T$ (2543 ± 14 µmol kg$^{-1}$) was observed in June. The reported salinity-$A_T$ relationship is thus
generated from monthly means ($R^2 = 0.74$) where $A_T$ units are µmol kg$^{-1}$ and error terms are SE:
$$A_T = 1554.9(\pm185.9) + 26.3(\pm4.9) \times S \qquad (2)$$


**3.4. High-frequency pH data**

To verify the weekly sampling scheme at Point B, a total of 11 SeaFET deployments
were conducted from June 2014 to April 2016, averaging 58 ± 25 days and 5 ± 2 calibration





samples per deployment (Fig. 6). Only 5 % of the data was removed during quality control, due
to biofouling in one deployment and a drained battery in another, yielding 610 days of good data.
The mean offset between calibration samples and the calibrated SeaFET pH time-series was ±
0.007 units $pH_T$, indicating a high-quality pH dataset (Fig. 6c). Sensor data corroborated the
seasonal pH and temperature cycle observed at Point B. Event-scale effects (e.g., $pH_T$ change $\geq$
0.1 for days to weeks, *sensu* Kapsenberg and Hofmann 2016) were absent at this site suggesting
that weekly sampling was sufficient to describe seasonal and interannual changes in carbonate
chemistry at Point B. Magnitude of diel $pH_T$ variability was small (the 2.5th to 97.5th percentiles
ranged between 0.01 and 0.05 units $pH_T$) and unrelated to seasonal warming or the concentration
of Chlorophyll-a (Fig. S2).

**4. Discussion**
**4.1. Observed changes in carbonate chemistry**

Anthropogenic forcing of $CO_2$ in seawater necessitates long-term monitoring, in order to

understand and study future ecological change in the coastal environment. In the coastal NW
Mediterranean Sea at EOL, near Point B, high-frequency pH data validated that weekly, morning
sampling at Point B was sufficient to capture water mass changes that were independent from
benthic, diel pH variability in seagrass beds inside the bay (Cox et al., 2016). Based on weekly
samples collected from 2007 through 2015, we detected anomaly changes in $pH_T$ of -0.0028 ±
0.0003 units $yr^{-1}$ and an increase in $A_T$ of 2.08 ± 0.19 µmol $kg^{-1}$ $yr^{-1}$. The corresponding $pCO_2$
increase of 3.53 ± 0.39 µatm $yr^{-1}$ is 70 % greater than the atmospheric $CO_2$ anomaly of +2.08 ±
0.01 ppm $yr^{-1}$ at Plateau Rosa, Italy. While Point B is a weak sink for $CO_2$ (De Carlo et al.,
2013), the increase in anthropogenic $CO_2$ in the atmosphere alone does not account for all of the



observed changes in carbonate chemistry at this location. The increase in $C_T$ was the dominant
driver of $pH_T$ change over the study period. An increase in $A_T$ partially buffered this $C_T$ increase
such that the observed acidification rate is only slightly larger than those reported at other ocean
time-series sites (-0.0026 to -0.0013 units $yr^{-1}$, Bates et al., 2014). Warming contributed to
approximately 35 % of the pH decline, which agrees well with the 30 % approximation at
DYFAMED, an open-sea site about 50 km offshore from Point B (Marcellin Yao et al., 2016). A
longer time-series of temperature paired with pH observations will be necessary to definitively
characterize this relationship. While anomalously warm summer temperatures in 2015 drove the
warming trends in this study, this region has warmed steadily since 1980 (Parravicini et al.,
2015) and this is observable at 50 m as well (Fig. S1). Finally, the co-evolution of $A_T$ and $C_T$
changes suggests that changes in $C_T$ are also due to increasing $A_T$, which includes bicarbonate
($HCO_3^-$) and carbonate ($CO_3^{2-}$) ions (Fig. 4). We assess the spatial extent of these trends and
discuss potential drivers.

Fastest rate increases in $A_T$ and $C_T$ occurred from May through July; the period of

seasonal transition (Fig. 3). At this time, various biophysical processes force the seasonality of
the carbonate system. In the NW Mediterranean, the main processes governing seasonal
variability in $A_T$ are evaporation increasing $A_T$ in summer (i.e., June through September at Point
B) and, to a lesser extent, phytoplankton uptake of nitrate ($NO_3^-$) and phosphate ($PO_4^{3-}$)
increasing $A_T$ from January through March (Cossarini et al., 2015). During the transition of these
processes, salinity decreases to a minimum in May, reflecting freshwater input that dilutes $A_T$ to
minimum values at the start of summer. For $C_T$, peak values occur in winter when the water
column is fully mixed (L. Mousseau, unpubl.). Mixing occurs down to more than 2000 m depth a
few 10s of km from the study site and $C_T$ is up to 100 µmol $kg^{-1}$ higher in deep waters (Copin-





Montégut and Bégovic, 2002). Following winter, $C_T$ declines due to a combination of
phytoplankton bloom carbon uptake and freshwater dilution, until the onset of summer
stratification. Summer warming leads to pCO$_2$ off-gassing to the atmosphere (De Carlo et al.,
2013), thereby further decreasing $C_T$. The increases in $A_T$ and $C_T$ from 2007 through 2015 were
more pronounced at 1 m compared to 50 m, suggesting that the driving processes dominate at the
surface.

To estimate if the observed trends from Point B were also occurring offshore, we turn to

time-series station DYFAMED, approx. 50 km from Point B (Fig. 1). The acidification rate at
DYFAMED was estimated at -0.003 ± 0.001 units pH yr$^{-1}$ from 1995 to 2011 (Marcellin Yao et
al., 2016), however, the uncertainty is large and makes the comparison with Point B unreliable.
$A_T$ at the DYFAMED did not change significantly during the period 2007-2014 ($F_{1,51}$ 3.204, $P =$
0.0794, $R^2$ 0.08, data from L. Coppola). This suggests that the processes driving changes in $A_T$
and $C_T$ at Point B, in addition to being shallow, dissipate offshore.

Another coastal area of the Mediterranean Sea, in the North Adriatic Sea, exhibited

similar changes in carbonate chemistry as those observed at Point B. By comparing cruise data
between the winters of 1983 and 2008, Luchetta et al. (2010) determined an acidification rate of
pH$_T$ -0.0025 units yr$^{-1}$ and an increase in $A_T$ of 2.98 µmol kg$^{-1}$ yr$^{-1}$ at depths < 75 m. As coastal
sites, Point B and the Gulf of Trieste in the N Adriatic Sea show a strong positive $A_{T0}$ (i.e.,
freshwater end-member alkalinity) in surface waters (Cantoni et al., 2012; Ingrosso et al., 2016).
In contrast to Point B, the N Adriatic Sea exhibits a negative salinity-$A_T$ relationship and faster
$A_T$ increase (Luchetta et al., 2010), suggesting that rivers may play a stronger role in $A_T$ trends
there compared to Point B. Rivers are significant sources of $C_T$ to the Gulf of Trieste, making up
3-16 % of $C_T$ in 2007 (Tamše et al., 2015). The authors note that 2007 was a year of record low



river discharge. Notably, this is the only year at Point B for which $A_{T0}$ was negative and salinity-
$A_T$ relationship was highly correlated, also indicating that 2007 was a year of low freshwater
input.
The correlates between Point B and N Adriatic Sea suggest a common driver of changes
in ocean carbonate chemistry at these two sites (possibly linked via shared watersheds of the
Alps), and these independent studies may by symptomatic of changes occurring across wider
coastal areas of the Mediterranean Sea. Monitoring efforts of carbonate chemistry in the eastern
Mediterranean Sea would offer an important contrast, as pH of eastern waters is expected to be
more sensitive to atmospheric $CO_2$ addition due to their ability to absorb more anthropogenic
$CO_2$ than either the western Mediterranean or Atlantic waters (Álvarez et al., 2014).

### 359      4.2. Potential drivers of changes in carbonate chemistry

Coastal ocean acidification rates vary greatly across different regions. In the NW Pacific
coast, rapid acidification of surface waters ($\Delta pH_T$ -0.058 units yr$^{-1}$) at Tatoosh Island has been
documented in the absence of concomitant changes in known drivers of local pH variability (e.g.,
upwelling, eutrophication, and more; Wootton and Pfister, 2012; Wootton et al., 2008). Further
inshore, in the Hood Canal sub-basin of the Puget Sound, only 24-49 % of the estimated pH
decline from pre-industrial values could be attributed to anthropogenic $CO_2$ (Feely et al., 2010).
The excess decrease in pH was attributed to increased remineralization (Feely et al., 2010).
Acidification rates documented along the North Sea Dutch coastline and inlets were highly
variable in space, with some exceeding the expected anthropogenic $CO_2$ rate by an order of
magnitude while others exhibited an increase in pH (Provoost et al., 2010). These sites
experience much greater sub-annual variability than Point B. As a coastal ocean acidification

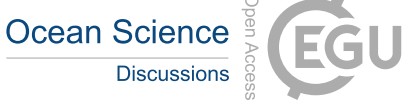

monitoring site, the relative simplicity of the Point B pH variability regime may therefore
provide an opportunity to further investigate the underlying drivers of rapid coastal ocean
acidification in the absence of additional noise from otherwise overshadowing processes.

While ocean acidification is often described in $pH_T$, at Point B, pH is largely a product of

underlying changes in $A_T$ and $C_T$. Sediment dissolution is unlikely to contribute to the observed
increase in $A_T$ as both aragonite and calcite were supersaturated throughout the study period;
conditions that are not conducive to large rates of dissolution. An overall reduction in calcium
carbonate ($CaCO_3$) precipitation rates is unlikely, as the dominant ecosystem in the Bay of
Villefranche-sur-Mer is seagrass meadows, which harbor relatively few calcifying organisms and
there has been no obvious changes in the abundance of calcifiers (J.-P. Gattuso, pers. obs.). All
the same, the sheer volume of the Bay would likely dilute any signature of changes in
calcification or dissolution of sediment or organisms. For the Mediterranean Sea, in general, the
influence of biogenic $CaCO_3$ on $A_T$ is small compared to the influence of river and Black Sea $A_T$
(Copin-Montégut, 1993). Increased input from the eastern Mediterranean Sea waters could
increase $A_T$, but this is questionable, because while eastern waters are higher in $A_T$, salinity and
pH are also greater and $C_T$ is a little lower, compared to the western waters (Álvarez et al., 2014;
Touratier and Goyet, 2011).

For $C_T$, there are a few additional drivers to discuss. First, anthropogenic $CO_2$ partly

accounted for the observed increase in $C_T$, independent of $A_T$. In summer, $pCO_2$ peaks and Point
B becomes a weak source of $CO_2$ to the atmosphere (De Carlo et al., 2013). As atmospheric $CO_2$
is increasing it may contribute to summertime $C_T$ retention, but this effect is likely too small for
the observed increases in $C_T$. Second, increased remineralization rates could contribute to $C_T$,
particularly in summer when the water is warmer, hence promoting respiration, and highly



stratified and isolated from $C_T$-rich bottom waters. However, this would require either (1) a
change in substrate (local or regional) for remineralization but no clear trend in particulate
organic carbon was observable in the study period (L. Mousseau, unpubl.), or (2) a change in net
community metabolism, but warming trends up through 2014 were not significant. Lastly,
Chlorophyll-a biomass, a proxy of primary production, has decreased since 1995 and blooms
have shifted towards earlier dates in the year, at Point B (Irisson et al., 2012). Both of these
processes could influence the increasing rate of $C_T$ in summer but would not account for the
increase in $A_T$.

The lack of salinity change excludes additional processes as drivers of carbonate

chemistry change at Point B. For example, increased summertime evaporation (concentration
effect) and reduced riverine discharge (decreased dilution effect) would both be expected to
cause an increase in salinity, which was not observed and so these are not suspected drivers.
However, changes in the content of freshwater sources is conceivable.

The observed changes in $A_T$ and $C_T$ could be achieved via augmented limestone

weathering increasing the $A_T$ input from land to the sea via rivers and submarine groundwater
springs. Increased $A_T$ in freshwater was documented in North American rivers (Raymond and
Cole, 2003; Stets et al., 2014) and groundwater (Macpherson et al., 2008). For the Mediterranean
Sea, rivers are a significant source of both $A_T$ and $C_T$ (Copin-Montégut, 1993). Riverine
contributions of $A_T$ originate from erosion and are correlated with bedrock composition (e.g.,
McGrath et al., 2016). The annual variability in salinity-$A_T$ relationships at Point B does suggest
influence of river discharge, as has been observed elsewhere in the Mediterranean Sea (Cantoni
et al., 2012; Turk et al., 2010). Signatures of limestone erosion can be observed in the $A_T$ of local
rivers near Point B (Var, Paillon, and Roya) and range between 1000 to 2000 μmol kg$^{-1}$ (data



from *Agence de l'Eau Rhône-Méditerranée-Corse*, http://sierm.eaurmc.fr). Even a rainwater
outfall at the entrance of Port de la Darse harbor, inside the Bay of Villefranche-sur-Mer, had an
$A_T$ of 607 ± 5 µmol kg$^{-1}$ (*N*=2) following a precipitation event. However, local precipitation was
not correlated with salinity or $A_T$, suggesting that rain runoff is not a driving factor of Point B
carbonate chemistry (Fig. S3). Lastly, submarine groundwater springs are a significant source of
nutrients, $A_T$, and $C_T$ to the ocean (Cai et al., 2003; Slomp and Van Cappellen, 2004). Submarine
springs have been identified along the Point B coastline (Gilli, 1995), but their carbonate
chemistry contributions are currently unknown.

Rivers as a potential driver of $A_T$ and $C_T$ trends at Point B could be achieved if the $A_T$

content discharged by rivers was changing, as was proposed for the Adriatic Sea (Luchetta et al.,
2010). For example, terrestrial organic matter cycling influences riverine $C_T$ (Vargas et al.,
2016), so changes in soil respiration could be expected to change $A_T$ of rivers. Increasing river $A_T$
has been documented in North America and occurs via a number of processes including: (1) the
interplay of rainfall and land-use (Raymond and Cole, 2003), (2) anthropogenic limestone
addition (a.k.a., liming) used to enhance agriculture soil pH (Oh and Raymond, 2006; Stets et al.,
2014) and freshwater pH (Clair and Hindar, 2005), and (3) potentially indirect effects of
anthropogenic $CO_2$ on groundwater $CO_2$-acidification and weathering (Macpherson et al., 2008).
Such, and other, processes are hypothesized to have driven $A_T$ changes in the Baltic Sea (Müller
et al., 2016). There, $A_T$ increased at a rate of +3.4 µmol kg$^{-1}$ yr$^{-1}$ during the period 1995 to 2014
(mean salinity = 7). In contrast to Point B, where salinity is about 38, the increase in Baltic Sea
$A_T$ was not noticeable at salinity > 30 (Müller et al., 2016). This contrast clouds the perspective
that changes in $A_T$ of freshwater sources can influence $A_T$ at Point B, and a more in depth study
will be necessary to address this.





Given the discussion above, the simplest plausible mechanisms causing changes in
carbonate chemistry at Point B would be through (1) increasing anthropogenic atmospheric $CO_2$,
and (2) increasing $A_T$ of freshwater sources (i.e., rivers, groundwater). Freshwater has a shallow
and coastal influence and is also dominant in the N Adriatic Sea, which exhibits similar trends as
those observed at Point B. If so, there is a lag effect, as freshwater influence peaks in May but $A_T$
and $C_T$ increased fastest from May through July. Consequently, this hypothesis needs further
investigation. The influence of a coastal boundary processes influencing seawater $A_T$ and $C_T$
presents a potentially major difference between coastal and offshore ocean acidification rates.
Until the source of $A_T$ increase is properly identified, use of this observation (e.g., in modeling)
should be implemented with great caution.

**5. Conclusion**
Predictions of coastal ocean acidification remain challenging due the complexity of
biogeochemical processes occurring at the ocean-land boundary and the lack of long-term
monitoring. At the Point B coastal monitoring station in the NW Mediterranean Sea, the ocean
acidification trend is greater than expected from assuming atmospheric equilibrium. We
postulate that the enhanced acidification trend could stem from changes in freshwater inputs
from land which are also the source of interannual variability in $A_T$ at this site. This study
highlights the importance of considering other anthropogenic influences in the greater land-sea
region that may (1) contribute to coastal biogeochemical cycles (*sensu* Duarte et al. 2013) and
(2) inform projections of anthropogenic change in near-shore waters and experimental design in
global ocean change biology.





**Data availability –** Time-series data from Point B are available at Pangaea[®] (doi:
10.1594/PANGAEA.727120)

**Author contribution –** JPG initiated the study, LM supervised data collection, SA performed
SeaFET deployments and calibration, JPG and LK designed and JPG conducted statistical
analyses, and LK prepared the manuscript with contributions from all authors.

**Competing interests -** The authors declare that they have no conflict of interest.

**Acknowledgements –** Thanks are due to the Service d'Observation Rade de Villefranche (SO-
Rade) of the Observatoire Océanologique and the Service d'Observation en Milieu Littoral
(SOMLIT/CNRS-INSU) for their kind permission to use the Point B data. Bottle samples were
analyzed for $C_T$ and $A_T$ by the *Service National d'Analyse des Paramètres Océaniques du CO2*.
Thanks are also due to Jean-Yves Carval, Anne-Marie Corre, Maïa Durozier, Ornella Passafiume
and Frank Petit for sampling assistance, to Alice Webb for her help with data analysis and to
Bernard Gentili and Jean-Olivier Irisson for producing Figure 1. Atmospheric $CO_2$ data from
Plateau Rosa was collected by Ricerca sul Sistema Energetico (RSE S.p.A.); we are grateful for
their contribution. The authors acknowledge L. Coppola for providing DYFAMED data and
Météo-France for supplying the meteorological data and the HyMeX database teams
(ESPRI/IPSL and SEDOO/Observatoire Midi-Pyrenees) for their help in accessing them. The
*Agence de l'Eau Rhône-Méditerranée-Corse* kindly provided data on the chemistry of local
rivers. Alexandre Dano, Gilles Dandec and Dominique Chassagne provided the high-resolution
bathymetric data for the volume estimate of the Bay. We are grateful for helpful comments from



Nicolas Metzl on the manuscript. This work is a contribution to the European Project on Ocean
Acidification (EPOCA; contract # 211384) and the MedSeA project (contract # 265103), which
received funding from the European Community's Seventh Framework Programme, and to the
United States National Science Foundation Ocean Sciences Postdoctoral Research Fellowship
(OCE-1521597) awarded to LK.

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





**Table 1.** Ocean acidification studies estimating or documenting pH change in the Mediterranean
Sea. TrOCA is the 'Tracer combining Oxygen, inorganic Carbon, and total Alkalinity' method,
N.R. means 'not reported', and PI is 'pre-industrial era'. *indicates studies where the reported
pH change was assumed to be at *in situ* temperatures.

| Region | Site | Method | Study period | pH scale | ºC | $\Delta pH\ yr^{-1} \pm SE$ | Total $\Delta pH$ | Reference |
|---|---|---|---|---|---|---|---|---|
| NW | Point B | time-series, anomaly | 2007-2015 | total | *in situ* | -0.0028 ± 0.0003 | -0.0252 | This study |
| NW | Point B | time-series, anomaly | 2007-2015 | total | 25 | -0.0017 ± 0.0002 | -0.0153 | This study |
| NW | Point B | model | 1967-2003 | total | *in situ* | -0.0014 | -0.05 | Howes et al. (2015) |
| NW | DYFAMED | time-series, observed | 1995-2011 | seawater | 17.34 | -0.003 ± 0.001 | -0.051 | Marcellin Yao et al. (2016) |
| NW | DYFAMED | time-series comparison | 1998-2000, 2003-2005 | seawater | *in situ** | - | -0.02 | Meier et al. (2014) |
| NW | Gulf of Lion | TrOCA | PI-2011 | NR | *in situ** | - | -0.15 to -0.11 | Touratier et al. (2016) |
| East | N Adriatic Sea | cruise comparison | 1983, 2008 | total | 25 | -0.0025 | -0.063 | Luchetta et al. (2010) |
| East | Otranto Strait | TrOCA | PI-1995 | seawater | 25 | - | < -0.1 to -0.05, ± 0.014 | Krasakopoulou et al. (2011) |
| Total | Full profile | TrOCA | PI-2001 | NR | *in situ** | - | -0.14 to -0.05 | Touratier and Goyet (2011) |
| Total | Bottom waters | model | 1800-2001 | total | *in situ** | - | -0.06 to -0.005 | Palmiéri et al. (2015) |
| Total | Surface waters | model | 1800-2001 | total | *in situ** | - | -0.084 ± 0.001 | Palmiéri et al. (2015) |
| Gibraltar Strait | Espartel sill | pH, pCO2 sensors | 2012-2015 | total | 25 | -0.0044 ± 0.00006 | - | Flecha et al. (2015) |





**Table 2.** Time-series (observed values and anomalies) regression analyses on seawater carbonate
chemistry, at Point B, 1 m, for salinity (S), temperature (T, ºC), dissolved inorganic carbon ($C_T$,
µmol kg$^{-1}$), total alkalinity ($A_T$, µmol kg$^{-1}$), pH$_T$, 25 ºC-normalized pH$_T$ (pH$_{T25}$), calcite ($\Omega_c$) and
aragonite ($\Omega_a$) saturation state, and salinity-normalized $A_T$ (n$A_T$) and $C_T$ (n$C_T$). Slope is change
yr$^{-1}$. $P \ll 0.001$ indicate p-values far smaller than 0.001.

|  | *Variable* | *Slope ± SE* | *Intercept ± SE* | *F* | *df* | *Slope P* | *R$^2$* |
|---|---|---|---|---|---|---|---|
|  | S | 0.0031 ± 0.0054 | 31.7 ± 10.8 | 0.326 | 1,415 | 0.568 | 0.001 |
|  | T | 0.12 ± 0.08 | -226 ± 157 | 2.429 | 1,411 | 0.12 | 0.006 |
|  | $C_T$ | 2.71 ± 0.31 | -3208 ± 626 | 75.671 | 1,414 | $\ll$0.001 | 0.155 |
|  | $A_T$ | 2.24 ± 0.21 | -1945 ± 418 | 115.895 | 1,415 | $\ll$0.001 | 0.218 |
|  | pH$_T$ | -0.0031 ± 0.0009 | 14.3 ± 1.8 | 11.674 | 1,410 | 0.001 | 0.028 |
| ***Observed*** | pH$_{T25}$ | -0.0012 ± 0.0004 | 10.5 ± 0.9 | 6.861 | 1,410 | 0.009 | 0.016 |
|  | pCO$_2$ | 3.78 ± 0.98 | -7214 ± 1976 | 14.82 | 1,410 | $\ll$0.001 | 0.035 |
|  | $\Omega_c$ | -0.0043 ± 0.0052 | 13.9 ± 10.5 | 0.684 | 1,410 | 0.409 | 0.002 |
|  | $\Omega_a$ | -0.0017 ± 0.004 | 6.74 ± 7.98 | 0.177 | 1,410 | 0.674 | 0 |
|  | n$A_T$ | 2.01 ± 0.31 | -1489 ± 632 | 41.067 | 1,410 | $\ll$0.001 | 0.091 |
|  | n$C_T$ | 2.57 ± 0.47 | -2918 ± 944 | 29.927 | 1,410 | $\ll$0.001 | 0.068 |
|  | S | -0.0017 ± 0.0044 | 3.38 ± 8.82 | 0.147 | 1,415 | 0.702 | 0 |
|  | T | 0.072 ± 0.022 | -145 ± 44 | 10.999 | 1,411 | 0.001 | 0.026 |
|  | $C_T$ | 2.97 ± 0.20 | -5965 ± 400 | 221.87 | 1,414 | $\ll$0.001 | 0.349 |
|  | $A_T$ | 2.08 ± 0.19 | -4189 ± 379 | 122.429 | 1,415 | $\ll$0.001 | 0.228 |
|  | pH$_T$ | -0.0028 ± 0.0003 | 5.72 ± 0.66 | 74.205 | 1,410 | $\ll$0.001 | 0.153 |
| ***Anomaly*** | pH$_{T25}$ | -0.0017 ± 0.0002 | 3.46 ± 0.43 | 64.204 | 1, 410 | $\ll$0.001 | 0.1354 |
|  | pCO$_2$ | 3.53 ± 0.39 | -7105 ± 776 | 83.927 | 1,410 | $\ll$0.001 | 0.17 |
|  | $\Omega_c$ | -0.0109 ± 0.0022 | 22.0 ± 4.5 | 24.08 | 1,410 | $\ll$0.001 | 0.055 |
|  | $\Omega_a$ | -0.0064 ± 0.0015 | 12.9 ± 3.1 | 17.33 | 1,410 | $\ll$0.001 | 0.041 |
|  | n$A_T$ | 2.20 ± 0.28 | -4425 ± 560 | 62.34 | 1,410 | $\ll$0.001 | 0.132 |
|  | n$C_T$ | 3.12 ± 0.29 | -6275 ± 579 | 117.486 | 1,410 | $\ll$0.001 | 0.223 |





**Table 3.** Contribution of temperature ($T$), salinity ($S$), total alkalinity ($A_T$), and dissolved
inorganic carbon ($C_T$) to observed changes in $pH_T$ and $pCO_2$ (µatm) $yr^{-1}$. The sum of the slopes
($\frac{dpH_T}{dt}$, $\frac{dpCO_2}{dt}$) is slightly inflated compared to the observed trends reported in Table 2 (-0.0037 vs. -
0.0031 $yr^{-1}$ for $pH_T$, and 4.26 vs. 3.78 µatm $yr^{-1}$ for $pCO_2$). These differences are negligible
relative to the error associated with the slope estimates. Incomplete sum of % contributions are
due to rounding. *P* of <<0.001 indicate p-values far smaller than 0.001.

| | Variable | Slope ± SE | % contribution | Slope P |
|---|---|---|---|---|
| | $T$ | -0.0013 ± 0.0009 | 35 | 0.15 |
| $\frac{\partial pH_T}{\partial var} \frac{dvar}{dt}$ | $S$ | -0.0002 ± 0.0008 | 5 | 0.793 |
| | $A_T$ | 0.0028 ± 0.0009 | -76 | 0.002 |
| | $C_T$ | -0.0050 ± 0.0009 | 135 | <<0.001 |
| $\frac{dpH_T}{dt}$ | | -0.0037 | 99 | |
| | $T$ | 1.43 ± 0.97 | 34 | 0.143 |
| $\frac{\partial pCO_2}{\partial var} \frac{dvar}{dt}$ | $S$ | 0.24 ± 0.89 | 6 | 0.792 |
| | $A_T$ | -2.94 ± 0.94 | -69 | 0.002 |
| | $C_T$ | 5.53 ± 0.96 | 130 | <<0.001 |
| $\frac{dpCO_2}{dt}$ | | 4.26 | 101 | |





**Figure 1.** Map of study region in the NW Mediterranean Sea (a) along the North current (b) in
the Bay of Villefranche-sur-Mer, France (c). Point B station, EOL buoy, and offshore time-series
station DYFAMED are marked. Bathymetric line units are m (c).

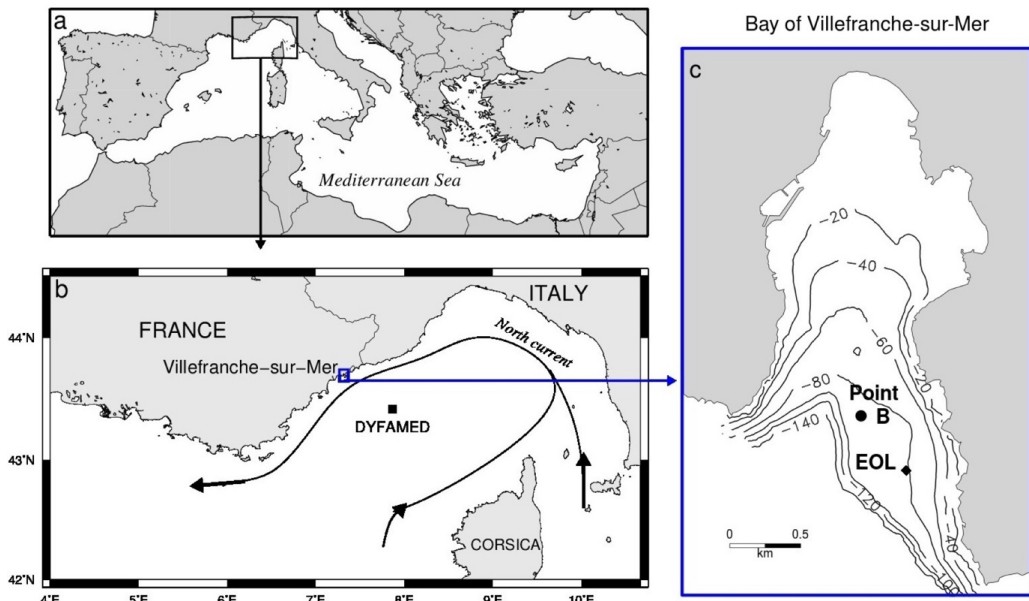




**Figure 2.** Time-series observations (a-g) and anomalies (h-n) of temperature, salinity, and

seawater carbonate chemistry at Point B, 1 m. Regression slopes are drawn ± SE (in grey) and

noted with a star for significance at α=0.05.

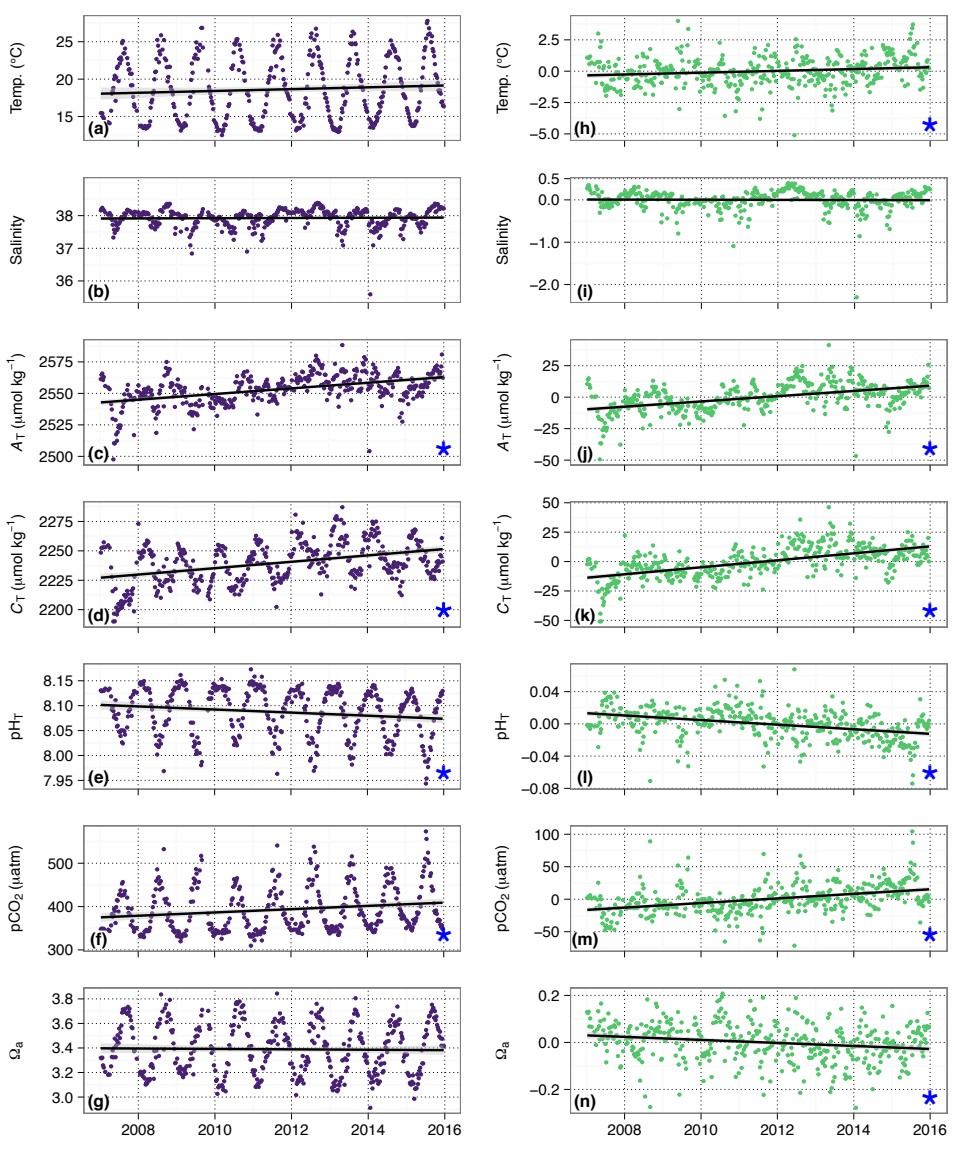





**Figure 3.** Monthly distribution of seawater carbonate chemistry at Point B, 1 m., using a
combination of a violin plot showing the relative frequency of the observations (shaded blue
area) and a boxplot showing the median, first and third quartiles, as well as outliers (blue).

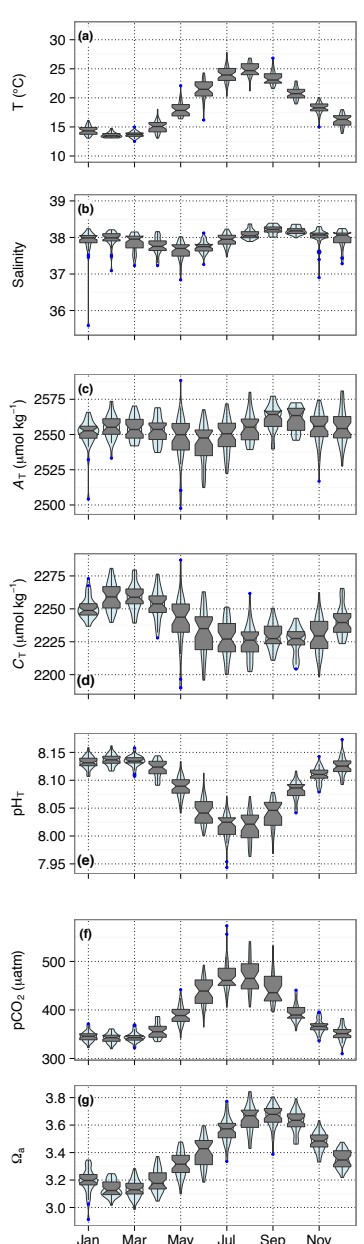




**Figure 4.** Increase in total alkalinity ($A_T$, purple) and dissolved inorganic carbon ($C_T$, green) by
month for the period 2007-2015. Errors bars are ± SE of the slope estimate and significance is
noted (*) at α=0.05.

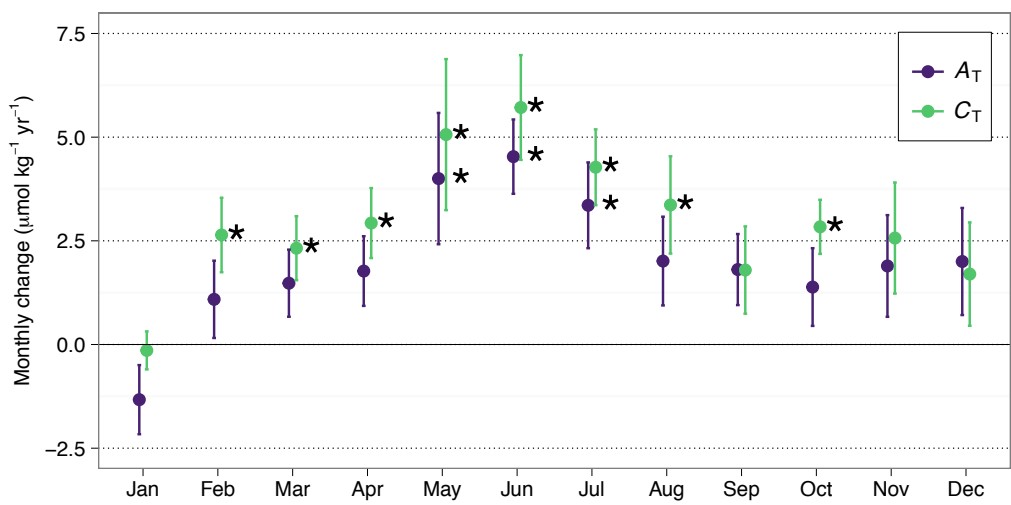






**Figure 5.** Salinity and total alkalinity relationships by year for the period 2007-2015, at Point B,
1 m. Data points are colored for month.

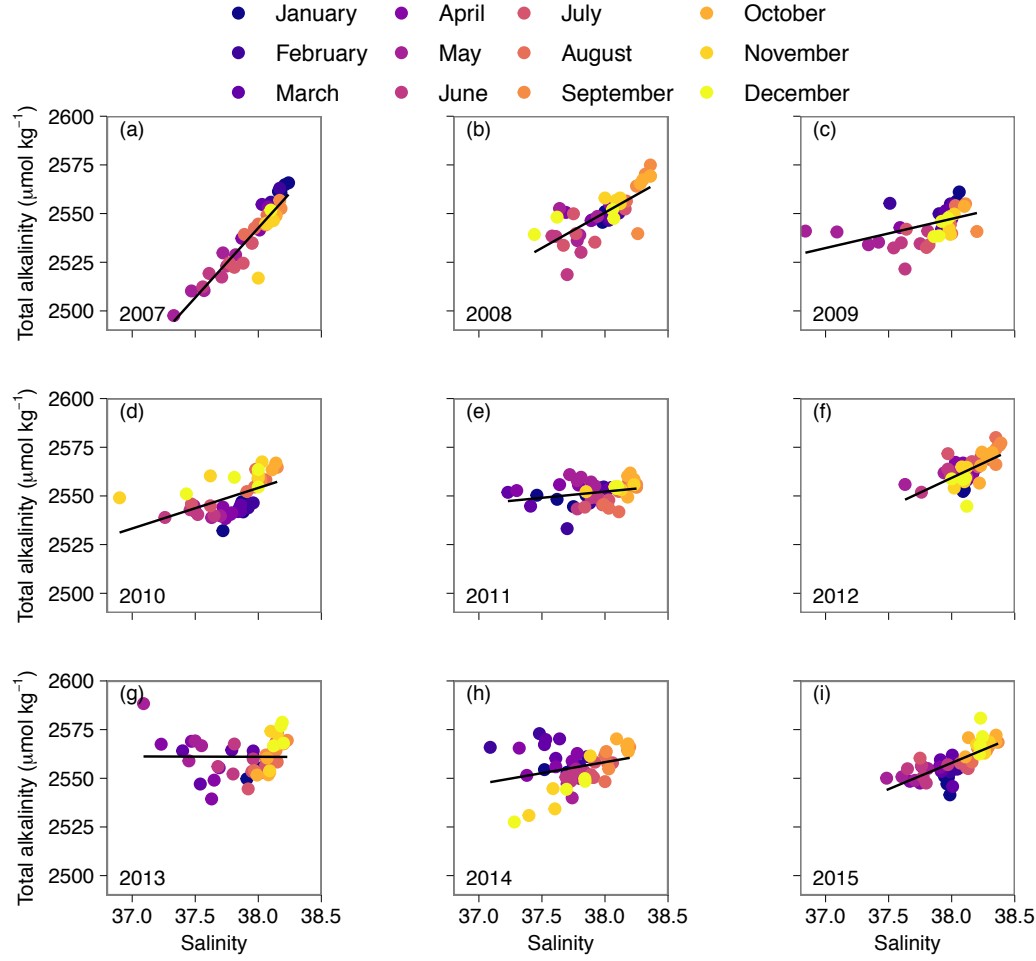






**Figure 6.** Time-series pH (a) and temperature (b) from autonomous SeaFET pH sensor at EOL
buoy, 2 m. Discrete calibration samples are noted in green, and grey vertical lines bracket
deployment periods (a). Mean offset of calibration samples from processed pH time-series was
$pH_T$ ±0.007 (c). Diel pH range was small and peaked in April and May (d) and exhibits a clear,
small, diel cycle (e, representative example from May 2015).

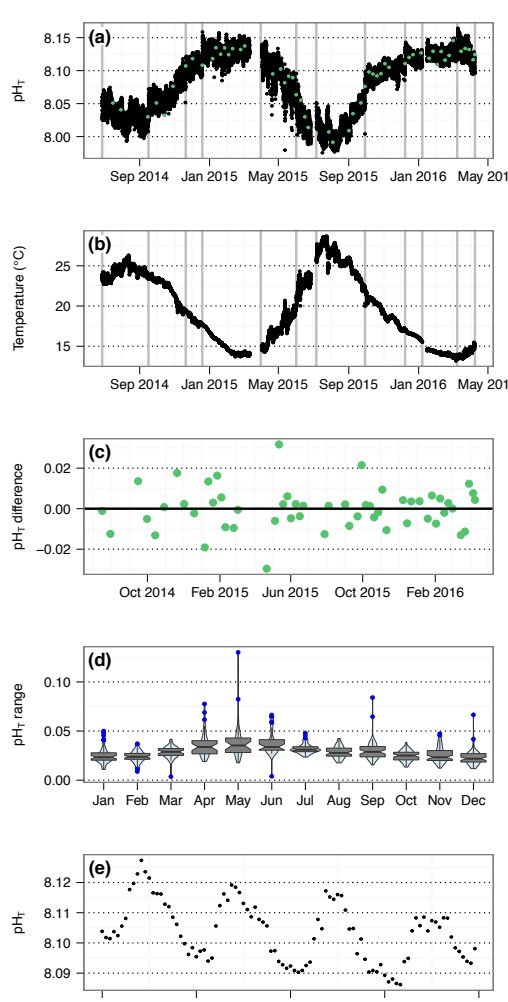
