# Peer review of "Coastal ocean acidification and increasing total alkalinity in the NW Mediterranean Sea"

_Ocean Science, 2016_

## Referee Comment (RC1) · Anonymous Referee #1 · 26 Oct 2016

In the present paper, the authors analyse two timeseries of carbonate system parameters (e.g. pH, alkalinity, CT and CO2) collected in a coastal site of the NW Mediterranean Sea during the period 2006-2015. A valuable description of the temporal variability at different scales and of recent trends is provided along with a discussion about the possible driving processes. The analysis presented in the paper provides important elements that can shed some light on the dynamics of carbonate system in coastal areas of the Mediterranean Sea. Therefore, the paper is worthy of being published in OS after few major concerns. Statistical methods are mostly appropriate, however the deconvolution analysis, the presence of two trend analyses and the relationship between atmospheric CO2 and sea water CO2 needs some clarifications (see major comments

2, 3 and 4). Results about Point B timeseries are well presented but I would suggest exploiting better the high-frequency timeseries at EOL boy (see major comments 5). Few points of the discussion section seem questionable and need some clarifications: that one about the relationship between the study site and the Adriatic Sea (major comment 6) and that one on the potential drivers (major comment 7). Finally, abstract and conclusion do not summarise exhaustively the valuable work and findings presented in the paper (point 1).

Major comments: 1) Abstract. I have found the abstract poorly informative, lacking to explain the main focus and the relevant findings. The first sentence is not clear to me. It is undoubtedly that monitoring in coastal area is important, however it seems to me that this sentence combines too many concepts. Please, review it. Line 18. The concept "faster-than-expected based on atmospheric carbon dioxide forcing alone", which is repeated twice (at lines 18 and 27), is not clear and needs some clarifications. Line 27. The sentence "localized biogeochemical cycling" should be made clearer. Line 22. The sentence "...its cause remains to be identified" is not consistent with the following "It seems therefore likely that changes in coastal AT cycling via a shallow coastal process gave rive to these observations". Please, review consistently. The sentence "Interesting, the increase . . ." (line 23) should be improved. Which "increases" is referred to? If the authors refer to the trends computed for each month, please make it clearer. Last sentence seems quite long and difficult to read. Please, rephrase it Keywords. The two keywords "global ocean change" and "near-shore" seems to me misleading. I would remove "global ocean change" since it is not a topic of the paper, and I would suggest using "coastal area" (as it used in the title) instead of "near shore" Conclusion. Few lines about the main findings of the analysis are maybe missing. This would help the paper to convey a clear take home message.

2) The deconvolution method (section 2.3) and results (section 3.2) should be revised. Authors should provide some details on their calculation method explaining how they deal with the hypotheses of linearity and of constant derivatives in time. These hypothe-

ses hold when dealing with annual values (according to Garcia-Ibanez et al. (2016)), but it seems they do not with the weekly data used in the present analysis. In fact, (as an example) the sum of the pH changes caused by the individual drivers differs of about 30% with respect to anomalies pH trends (21% to observed pH trends). It is said that these differences are negligible (line 774), however no clear explanation is given. A comment on this issue should be provided.

3) The authors state that the timeseries is "detrended for seasonality by subtracting monthly mean . . . resulting anomalies were analysed using a linear regression" (Lines 160-164). To my understanding the analysis is performed according to the approach provided by Bates et al., (2014). Given that, why is the linear regression computed on observations in addition to that one on anomalies? These two trend estimates (one on anomalies and one on observations, Table 2) are slightly different but no explanation or discussion is provided but they are used in different part of the text (i.e. the trends on anomalies are commented throughout the text, while the trends on observations are used in deconvolution analysis). This might be misleading. Author should decide which type of timeseries model they are proposing (i.e. first trend then seasonality or vice versa) and use only one.

4) Line 248-251. Not clear. Do the authors propose a linear model relationship between atmosphere $CO_2$ trend and seawater $CO_2$ trend? This should be clarified as well as the assumption of air-sea $CO_2$ equilibrium. As a consequence, the discussion at line 299-300 seems not well supported by the result. A comment on this issue should be provided since it is claimed that this is one of the most important drivers (see lines 440-442 in discussion).

5) Section 3.4. EOL time series is very interesting and it could be better exploited. In particular, I would suggest that the search for event-scale effects should be made considering the variability of pH at local scale (and not using a threshold which is valid for open ocean). In fact, plot 6d shows the presence of daily pH variations larger than 0.05. These possible event-scale effects could be investigated. Most importantly,

authors could resolve the pH variability at daily, seasonal, events and interannual temporal scales producing an additional interesting result. I would encourage the authors to exploit this time series not only for validating the weekly one.

6) Discussion at lines 338-357. It is not convincing the claimed relationship between Point B and the Northern Adriatic Sea. While the Adriatic Sea has a negative relationship with salinity, Point B has a positive relationship with salinity (eq. 2 at line 274). Therefore, the comparison between the two sites is poorly informative of the behaviour of carbonate system at Point B. I would suggest reducing this part of the discussion to those elements that help in understanding Point B dynamics.

7) Discussion about the drivers of AT and CT trends (lines 376-449). This part, although very interesting, is maybe too long and sometimes not well connected to the results. I would suggest shortening this part, focusing on those drivers that are thought to play the most relevant role. Since the authors claim that terrestrial input are important, a description of the rivers and underground sources in the region of the Bay of Villefranche-sur-Mer (and their contributions to AT and CT) should be added into the introduction and used in the discussion for inferring the changes required to explain the observed trends. Moreover, a budget of the AT and CT for the Bay could be estimated, considering the volume of the bay, the exchanges with open sea, and the input terms (from atmosphere and terrestrial sources). This analysis can shed some light on the relative importance of the different boundaries to explain the observed trends, and , eventually, quantifying the missing term.

Minor points: Lines 107 and 128. Please provide the exact length of the timeseries.

Line 118. A description of the riverine input in the area could be of interest. Have rivers along the coast near the Bay of Villefranche-sur-Mer high AT (lines 118-119) or low AT (line 122)?

Line 215. Not clear what "exception" the authors refer to.

Lines 222-223. Why do the authors report that the T trend on anomalies is not significant for the period 1999-2014? Removing the last year (which has high T) seems a subjective choice that should be clarified. If, for any reason, the year 2015 is considered an outlier and it has to be removed, it should be done for all the variables.

Line 231-234. The sentence is quite long and difficult to read, please rephrase it.

Line 254. Does "which peaked in June" refer to parameters or to their monthly trends?

Section 3.3. Lines 262 and 274. It is not clear the message that the authors want to convey. At line 262 it is said that salinity is a poor predictor of AT, however the section ends with a salinity-alkalinity regression. Please, review this section consistently.

Lines 294-295. The sentence is long and difficult to read. Please rephrase it. Further, what do the authors mean for "morning sampling"? Is it referred to the sampling procedure of the Point B timeseries? If so, it should be introduced in Material&Method, and motivation explained if important.

Line 311-312. This sentence seems inaccurate. Which is the causal factor of CT increase due to AT increase?

Line 313. Which "spatial extent" do the authors mean?

Lines 336-337. The analysis of the coastal–offshore gradient would deserve some addition investigations, since offshore deep water is supposed to play a role for CT evolution at point B (at lines 323-325).

Lines 359-369. This part could be moved to introduction.

Lines 772-776. Table 3 caption reports not only the description of the table but also comments on results. Please remove the no necessary text.

Lines 799-801. Figure 6. Please use the caption to describe the plots without describing the results.

---

## Referee Comment (RC2) · Anonymous Referee #2 · 22 Nov 2016

Interactive comment on "Concomitant ocean acidification and increasing total alkalinity at a coastal site in the NW Mediterranean Sea (2007 -2015) by Lydia Kapsenberg et al. Anonymous Referee #2

The article presents, analyses, and discusses the time series of physical parameters and carbonate system properties gathered in a coastal station of the NW Mediterranean Sea and spanning nearly one decade. The analyses on different time scales of ocean acidification, responsible for changes in the marine CO2 system with effects on the dissolved inorganic carbon (CT), partial pressure of CO2 (pCO2), pHT, total alkalinity (AT) and calcium carbonate saturation states is valuable. Actually, sustained observations of inorganic carbon parameters by means of long term time series count

on a few sites over the global oceans, and on even fewer in the Mediterranean basin, especially in the coastal area, although such shallow zones can be exposed to intense land sea interactions and to a great complexity of physical and biological processes interacting with ocean acidification. Ocean acidification in coastal zones is remarkably difficult to predict. The present paper can contribute to a better knowledge of the coastal systems vulnerability to ocean acidification, by investigating on the multiple drivers eventually working in this environment. In particular, the analysis of the time series trends (§3.1) appears robust and the decomposition of ocean acidification into the principal drivers is appropriate and informative. The method applied for the deconvolution of pHT and pCO2, (proposed in §3.2 and 3.3,) is new for both the Med Sea and coastal regions, and provides useful indications on different processes driving ocean acidification in this site in comparison with open ocean. The discussion of CT and AT increases as the main drivers of pH decrease (§4.1 and 4.2 ) is well conducted. The proposed attribution of these two concomitant increases (AT and CT) to terrestrial inputs with changing and increasing AT and CT (riverine water and/or to groundwater springs) seems reasonable. The hypothesis is deeply discussed, although not enough supported by the correlation with low salinity (see major comments). Finally, conclusions well enhance the role of land sea interactions. In conclusion, the publication of this article in OS is worth after a few revisions (reported below).

Major Comments: 1. Discussion on drivers of AT and CT trends (line 402-439). I agree with the suggestion of Referee #1. The description (in terms of carbonate chemistry) of the river and underground sources in the region of the Bay of Villefranche should be added for a more complete discussion.. 2. Then I'm wondering whether results (§3.3) and discussion (§4.2) might benefit from the recent paper by Fry et al. (2015). In the paper authors calculate "Alk1", the same as normalized AT, to remove the contribution of evaporation and precipitation, and calculate "Alk2" (accounting for river AT) to remove the riverine input. This was done because Friis et al. (2003) found that misleading results are produced if normalized alkalinity is used in ocean regions receiving river outflows. . .. It seems to me that you have all the necessary data (Alkm= measured

alkalinity, salinity and Alkr=river alkalinity) requested to calculate Alk2. Following the subtraction of these major processes that affects alkalinity at Point B, I would expect: salinity- AT relationship (through the 9 years of time series) improves the positive AT anomaly disappears (or at least decreases)

C.H. Fry, T. Tyrell, M.P. Hain, N.R. Bates and E.P. Achtenberg. Analysis of global surface ocean alkalinity to determine controlling processes. Marine Chemistry, 174,46-457, 2015. Friis K., Koertzinger A. and Wallace D.W.R., 2003. The salinity normalization of marine inorganic carbon chemistry data. Geophys. Res. Lett. 30 (2), 1085.

Minor comments: Lines 222-224. Temperature anomaly increased but this significance was lost with the exclusion of the year 2015. . .. The sentence is not clear to me. Do you mean that excluding the year 2015 (it was exceptionally warm and SST raised to the highest values during summer) there was no increase in temperature anomaly ? and therefore the temperature increase cannot be the driver of observed changes in the carbonate system properties ? In any case please rephrase the sentence. Lines 320-322. During the transition of these processes, salinity decreases to a minimum in May, reflecting the freshwater input that dilutes AT to minimum values. . . Fresh water is able to dilute AT to minimum values if discharging rivers have lower AT than seawater but this might be not true in case of rivers draining carbonatic watershed (later, line 416, authors report that outflowing rivers into the Bay of Villefranche have high AT). This can be misleading, please modify. Line 325-326. Following winter CT declines due to a combination of phytoplankton bloom carbon uptake and freshwater dilution. . . Again I would be more cautious as freshwater dilutes if it contains lower CT . . . Lines 351-357. The correlates between Point B and N Adriatic Sea suggest a common driver of changes in ocean carbonate chemistry at these two sites (possibly linked via shared watersheds of the Alps) . . . ..... Mediterranena Sea. Could you explain and present more clearly this hypothesis ?

Please also note the supplement to this comment:

http://www.ocean-sci-discuss.net/os-2016-71/os-2016-71-RC2-supplement.pdf

**Supplement:**

The article presents, analyses, and discusses the time series of physical parameters and carbonate system properties gathered in a coastal station of the NW Mediterranean Sea and spanning nearly one decade.

The analyses on different time scales of ocean acidification, responsible for changes in the marine $CO_2$ system with effects on the dissolved inorganic carbon ($C_T$), partial pressure of $CO_2$ ($pCO_2$), $pH_T$, total alkalinity ($A_T$) and calcium carbonate saturation states is valuable. Actually, sustained observations of inorganic carbon parameters by means of long term time series count on a few sites over the global oceans, and on even fewer in the Mediterranean basin, especially in the coastal area, although such shallow zones can be exposed to intense land sea interactions and to a great complexity of physical and biological processes interacting with ocean acidification. Ocean acidification in coastal zones is remarkably difficult to predict. The present paper can contribute to a better knowledge of the coastal systems vulnerability to ocean acidification, by investigating on the multiple drivers eventually working in this environment.

In particular, the analysis of the time series trends (§ 3.1) appears robust and the decomposition of ocean acidification into the principal drivers is appropriate and informative. The method applied for the deconvolution of $pH_T$ and $pCO_2$, (proposed in §3.2 and 3.3,) is new for both the Med Sea and coastal regions, and provides useful indications on different processes driving ocean acidification in this site in comparison with open ocean.

The discussion of $C_T$ and $A_T$ increases as the main drivers of pH decrease (§ 4.1 and 4.2 ) is well conducted. The proposed attribution of these two concomitant increases ($A_T$ and $C_T$) to terrestrial inputs with changing and increasing $A_T$ and $C_T$ (riverine water and/or to groundwater springs) seems reasonable. The hypothesis is deeply discussed, although not enough supported by the correlation with low salinity (see major comments).

Finally, conclusions well enhance the role of land sea interactions.

In conclusion, the publication of this article in OS is worth after a few revisions (reported below).

Major Comments:

1. Discussion on drivers of $A_T$ and $C_T$ trends (line 402-439). I agree with the suggestion of Referee #1: a description (in terms of carbonate chemistry) of the river and underground sources in the region of the Bay of Villefranche should be added for a more complete discussion..

2. Then I'm wondering whether results (§ 3.3) and discussion (§ 4.2) might benefit from the recent paper by Fry et al. (2015). In the paper authors calculate "Alk$_1$", the same as normalized $A_T$, to remove the contribution of evaporation and precipitation, and calculate "Alk$_2$" (accounting for river $A_T$) to remove the riverine input. This was done because Friis et al. (2003) found that misleading results are produced if normalized alkalinity is used in ocean regions receiving river outflows….
   It seems to me that you have all the necessary data (Alk$_m$= measured alkalinity, salinity and Alk$_r$= river alkalinity) requested to calculate Alk$_2$.
   Following the subtraction of these major processes that affects alkalinity at Point B, I would expect: salinity- *$A_T$* relationship (through the 9 years of time series)  improves
   the positive $A_T$ anomaly disappears (or at least decreases)

   C.H. Fry, T. Tyrell, M.P. Hain, N.R. Bates and E.P. Achtenberg. Analysis of global surface ocean alkalinity to determine controlling processes. Marine Chemistry, 174,46-457, 2015.
   Friis K., Koertzinger A. and Wallace D.W.R., 2003. The salinity normalization of marine inorganic carbon chemistry data. Geophys. Res. Lett. 30 (2), 1085.

Minor comments:

Lines 222-224.  *Temperature anomaly increased but this significance was lost with the exclusion of the year 2015….*
The sentence is not clear to me. Do you mean that excluding the year 2015 (it was exceptionally warm and SST raised to the highest values during summer) there was no increase in temperature anomaly ? and therefore the temperature increase cannot be the driver of observed changes in the carbonate system properties ?
In any case please rephrase the sentence.

Lines 320-322.  *During the transition of these processes, salinity decreases to a minimum in May, reflecting the freshwater input that dilutes $A_T$ to minimum values…*
Fresh water is able to dilute $A_T$ to minimum values if discharging rivers have lower $A_T$ than seawater but this might be not true in case rivers drains carbonatic watershed (later, line 416, authors report that outflowing rivers into the Bay of Villefranche have high $A_T$).
This can be misleading, please modify.

Line 325-326.  *Following winter $C_T$ declines due to a combination of phytoplankton bloom carbon uptake and freshwater dilution…*
Again I would be more cautious as freshwater dilutes if it contains lower $C_T$ …

Lines 351-357.  *The correlates between Point B and N Adriatic Sea suggest a common driver of changes in ocean carbonate chemistry at these two sites (possibly linked via shared watersheds of the Alps) … ….. Mediterranena Sea.*
Could you explain and present more clearly this hypothesis ?

---

## Author Comment (AC1) · 16 Jan 2017

*Overall author response: We thank the referees for their time and suggested improvements. We appreciate the positive reception of our study and have addressed all major comments below, point by point. Our major revisions include:*

*1. Removal of trend analyses on raw observations. We now only present statistical analyses on anomalies.*

*2. Use of the anomaly approach for the deconvolution analyses. This has resolved issues raised by the Referees and we were better able to attribute trends to specific drivers of change (e.g., warming, atmospheric $CO_2$). As a result, the Abstract, Results, and Discussion have been updated.*

*For the ease of reviewing our response, we have numbered the Referee Comments (RC) from 1 to 27. RC#1-21 are those from Referee #1 and RC#22-27 are those from Referee #2. We first list the comment in standard font and then our response followed by the change in the text, in italics.*
* * *
**Anonymous Referee #1,**

In the present paper, the authors analyse two timeseries of carbonate system parameters (e.g. pH, alkalinity, CT and CO2) collected in a coastal site of the NW Mediterranean Sea during the period 2006-2015. A valuable description of the temporal variability at different scales and of recent trends is provided along with a discussion about the possible driving processes. The analysis presented in the paper provides important elements that can shed some light on the dynamics of carbonate system in coastal areas of the Mediterranean Sea. Therefore, the paper is worthy of being published in OS after few major concerns. Statistical methods are mostly appropriate, however the deconvolution analysis, the presence of two trend analyses and the relationship between atmospheric CO2 and sea water CO2 needs some clarifications (see major comments 2, 3 and 4). Results about Point B timeseries are well presented but I would suggest exploiting better the high-frequency timeseries at EOL boy (see major comments 5). Few points of the discussion section seem questionable and need some clarifications: that one about the relationship between the study site and the Adriatic Sea (major comment 6) and that one on the potential drivers (major comment 7). Finally, abstract and conclusion do not summarise exhaustively the valuable work and findings presented in the paper (point 1). Major comments:

*No response needed.*

1) Abstract. I have found the abstract poorly informative, lacking to explain the main focus and the relevant findings. The first sentence is not clear to me. It is undoubtedly that monitoring in coastal area is important, however it seems to me that this sentence combines too many concepts. Please, review it.

*We have extensively altered the abstract given the comments described below and updated results from an improved deconvolution analysis (see RC#2). The first sentence has been simplified and now reads, "Coastal time-series of ocean carbonate chemistry are critical for understanding how global anthropogenic change manifests in near-shore ecosystems".*

-Line 18. The concept "faster-than-expected based on atmospheric carbon dioxide forcing alone", which is repeated twice (at lines 18 and 27), is not clear and needs some clarifications.

*As part of the updated results presented in the abstract, we have removed this phrasing. Instead, we now directly describe the drivers of the observed trend in pH, which are atmospheric carbon dioxide and warming.*

-Line 27. The sentence "localized biogeochemical cycling" should be made clearer. Line 22. The sentence ". . .its cause remains to be identified" is not consistent with the following "It seems therefore likely that changes in coastal AT cycling via a shallow coastal process gave rise to these observations". Please, review consistently.

*We have clarified the sentence with "localized biogeochemical cycling" to specify the "...importance of understanding changes in coastal carbonate chemistry through the lens of biogeochemical cycling at the land-sea interface."*

*The sentence "its cause remains to be identified" has been updated to reflect that our dataset cannot identify the driver of $A_T$ change, but we have some information as what it could be. These concepts which were previously separated in the abstract but are now combined: "The driving process of the interannual increase in $A_T$ remains to be identified, but it has a seasonal and shallow component, which may indicate riverine or groundwater influence."*

-The sentence "Interesting, the increase . . ." (line 23) should be improved. Which "increases" is referred to? If the authors refer to the trends computed for each month, please make it clearer.

*We have clarified the statement by specifying reference to "monthly means". The edited sentence now reads: "Based on the analysis of monthly trends, concomitant increases in $C_T$ and $A_T$ were fastest in the spring-summer transition."*

-Last sentence seems quite long and difficult to read. Please, rephrase it.

*We have revised the end of the abstract.*

-Keywords. The two keywords "global ocean change" and "near-shore" seems to me misleading. I would remove "global ocean change" since it is not a topic of the paper, and I would suggest using "coastal area" (as it used in the title) instead of "near shore".

*We have removed "near-shore" and replaced "global ocean change" with "ocean acidification". To highlight the coastal aspect of our study, we have edited the title to lead with "coastal": "Coastal ocean acidification and increasing total alkalinity in the NW Mediterranean Sea".*

Conclusion. Few lines about the main findings of the analysis are maybe missing. This would help the paper to convey a clear take home message.

*We have expanded the main findings and take home message: "This study exemplifies the importance of understanding changes in coastal carbonate chemistry through the lens of biogeochemical cycling at the land-sea interface. This is the first coastal acidification time-series providing data at high temporal resolution. The data confirm rapid warming in the Mediterranean Sea and demonstrate coastal acidification with a synchronous increase in total alkalinity."*

2) The deconvolution method (section 2.3) and results (section 3.2) should be revised. Authors should provide some details on their calculation method explaining how they deal with the hypotheses of linearity and of constant derivatives in time. These hypotheses hold when dealing with annual values (according to Garcia-Ibanez et al. (2016)), but it seems they do not with the weekly data used in the present analysis. In fact, (as an example) the sum of the pH changes caused by the individual drivers differs of about 30% with respect to anomalies pH trends (21% to observed pH trends). It is said that these differences are negligible (line 774), however no clear explanation is given. A comment on this issue should be provided.

*We have now analyzed the regression on the deconvoluted time-series via the anomaly approach as used for generating the time-series trends (e.g., generating a time-series anomaly via subtraction of the respective monthly means). This revision improves the consistency of analyses presented throughout the paper (RC#3) and resolves the inconsistency between the sum of pH changes from the deconvolution compared to the reported trends (which were based on the raw data in the original version of the manuscript).*

*Using the anomaly approach, for pH, the sum of the pH changes generated by the deconvolution equals the observed change in pH of -0.0028 per year. For $pCO_2$, the sum of the slopes slightly underestimates the anomaly trend (6% difference) but falls within the range of error. We therefore conclude that the assumptions involved in the approach of Garcia-Ibanez et al. (2016) hold for our dataset. We have updated the methods and results to reflect this change in analysis.*

3) The authors state that the timeseries is "detrended for seasonality by subtracting monthly mean . . . resulting anomalies were analysed using a linear regression" (Lines 160-164). To my understanding the analysis is performed according to the approach provided by Bates et al., (2014). Given that, why is the linear regression computed on observations in addition to that one on anomalies? These two trend estimates (one on anomalies and one on observations, Table 2) are slightly different but no explanation or discussion is provided but they are used in different part of the text (i.e. the trends on anomalies are commented throughout the text, while the trends on observations are used in deconvolution analysis). This might be misleading. Author should decide which type of timeseries model they are proposing (i.e. first trend then seasonality or vice versa) and use only one.

*We used the same methods as Bates et al. (2014) in order to be able to compare our data and observed trends with those observed elsewhere and presented in Bates et al. (2014). Following Bates et al., 2014, we present both the raw observations and anomaly data. Presentation of the observation data visually shows the strong seasonality of carbonate chemistry parameters at this location and the raw values. We find this visualization useful and have therefore kept both the observed and anomaly data in Figure 2. We however removed analyses performed on the raw observations to avoid confusion (removal of regression lines from raw time-series observations in Fig. 2 and Fig. S1 of the original manuscript, and removal of regression results from statistics presentation in Table 2 and Table S1 of the original manuscript). Since this has simplified the presentation of the results, we have moved the analyses performed on data from 50 m, which were previously in the supplemental data, to the main document.*

*We have added an explanatory statement (Section 2.2.), which justifies the anomaly approach for comparative purposes with trends published in Bates et al. (2014).*

*In addition, we have adjusted the deconvolution to display anomaly trends to match the presentation of the time-series results (see RC #2). The deconvolution is also performed on data from 50 m, to maintain consistency with the presentation of time-series at both depths.*

4) Line 248-251. Not clear. Do the authors propose a linear model relationship between atmosphere CO2 trend and seawater CO2 trend? This should be clarified as well as the assumption of air-sea CO2 equilibrium. As a consequence, the discussion at line 299-300 seems not well supported by the result. A comment on this issue should be provided since it is claimed that this is one of the most important drivers (see lines 440-442 in discussion).

*The contribution of atmospheric CO$_2$ to the observed trends at Point B is necessary to discuss as it is the main driver of global ocean acidification. The deconvolution of pCO$_2$ allowed us to compare the rates of atmospheric and seawater CO$_2$ increase. By assuming a linear model relationship between atmospheric and seawater CO$_2$ we can estimate the potential maximum contribution of anthropogenic atmospheric CO$_2$ to the observed trends at Point B.*

*We have edited the text in Section 3.2 (Deconvolution results) to clarify this and describe the assumption we are making: "To estimate the maximum influence of anthropogenic CO$_2$ forcing at Point B, we assume air-sea CO$_2$ equilibrium (e.g., increase in atmospheric CO$_2$ causes an equal increase in seawater pCO$_2$). Considering the error associated with deconvolution of pCO$_2$ at 1 m, atmospheric CO$_2$ increase can, at most, represent 37-42 % of the total C$_T$ contribution $(\frac{\partial pCO_2}{\partial C_T}\frac{dC_T}{dt})$ to $\frac{dpCO_2}{dt}$. "*

5) Section 3.4. EOL time series is very interesting and it could be better exploited. In particular, I would suggest that the search for event-scale effects should be made considering the variability of pH at local scale (and not using a threshold which is valid for open ocean). In fact, plot 6d shows the presence of daily pH variations larger than 0.05. These possible event-scale effects could be investigated. Most importantly, authors could resolve the pH variability at daily, seasonal, events and interannual temporal scales producing an additional interesting result. I would encourage the authors to exploit this time series not only for validating the weekly one.

*We agree that the SeaFET time-series offers a unique opportunity to explore pH variability at a local and high-frequency scale, however, extensive discussion on this is outside the scope of this paper. Our data presentation is primarily focused on inter-annual changes and how these observations compare to other coastal and open ocean sites. In this light, we restrict the comparison of the high-frequency pH data to previously published categories in other coastal study regions (e.g., Kapsenberg et al., 2016 as mentioned in the manuscript). For example, under this perspective, at Point B event-scale pH variations defined by day to week(s)-long change in mean pH of ~0.1 are absent. We briefly touch on daily, events, and seasonal pH variation in the Results (Section 3.4). In addition, identifying drivers of the extremely small variation in daily pH variability would require other high-frequency data such as oxygen time-series or wind speed (as done in Kapsenberg et al., 2016). In this light, we have not altered the presentation of the EOL pH time-series.*

6) Discussion at lines 338-357. It is not convincing the claimed relationship between Point B and the Northern Adriatic Sea. While the Adriatic Sea has a negative relationship with salinity, Point B has a positive relationship with salinity (eq. 2 at line 274). Therefore, the comparison between the two sites is poorly informative of the behaviour of carbonate system at Point B. I would

suggest reducing this part of the discussion to those elements that help in understanding Point B dynamics.

*We have altered this paragraph to only present the results of the Northern Adriatic Sea and removed qualitative comparisons between the sites. Given the similar pH and $A_T$ changes, we now only state: "Point B and Adriatic Sea observations are independent but reflect changes in seawater chemistry that may be occurring across a wider coastal region."*

7) Discussion about the drivers of AT and CT trends (lines 376-449). This part, although very interesting, is maybe too long and sometimes not well connected to the results. I would suggest shortening this part, focusing on those drivers that are thought to play the most relevant role.

Since the authors claim that terrestrial input are important, a description of the rivers and underground sources in the region of the Bay of Villefranche-sur-Mer (and their contributions to AT and CT) should be added into the introduction and used in the discussion for inferring the changes required to explain the observed trends. Moreover, a budget of the AT and CT for the Bay could be estimated, considering the volume of the bay, the exchanges with open sea, and the input terms (from atmosphere and terrestrial sources). This analysis can shed some light on the relative importance of the different boundaries to explain the observed trends, and, eventually, quantifying the missing term.

*We have shortened and streamlined the Discussion Section 4.2. on potential drivers of $A_T$ and $C_T$ trends. Due to the lack of data on freshwater contributions to Point B we are unable to create a $A_T$ and $C_T$ budget. There is no river mouth next to Point B (closest rivers are 4, 10, and 25 km away which we have now described in the Methods, see RC#9) and contributions of groundwater are unknown. Likewise, residence time of water in the Bay and exchange with the open sea is not known. Unfortunately, we do not have the data to expand the discussion of $A_T$ and $C_T$ beyond what we have already presented.*

Minor points:

8) Lines 107 and 128. Please provide the exact length of the timeseries.

*We have added the length of the time-series in both locations.*

9) Line 118. A description of the riverine input in the area could be of interest. Have rivers along the coast near the Bay of Villefranche-sur-Mer high AT (lines 118-119) or low AT (line 122)?

*Line 118-119 (original submission) indicated that $A_T$ for rivers draining into the Mediterranean are generally high (for readers unfamiliar with this region) and line 122 was an example of that. We have moved the following sentence from the Discussion to this section in the methods to help clarify this: "Limestone erosion can be observed in the $A_T$ of rivers nearest to Point B (Paillon, due 4 km West; Var due 10 km West; and Roya due 26 km East). River $A_T$ ranges between 1000 to 2000 μmol $kg^{-1}$ (data from Agence de l'Eau Rhône-Méditerranée-Corse, http://sierm.eaurmc.fr), and is lower than seawater $A_T$." This sentence was also edited in response to RC#25.*

10) Line 215. Not clear what "exception" the authors refer to.

*The exception is that salinity and temperature were not changing faster at 1 m compared to 50 m. We have revised the sentence to read "carbonate chemistry parameters" instead of "carbonate chemistry", which a reader may not include salinity or temperature in.*

11) Lines 222-223. Why do the authors report that the T trend on anomalies is not significant for the period 1999-2014? Removing the last year (which has high T) seems a subjective choice that should be clarified. If, for any reason, the year 2015 is considered an outlier and it has to be removed, it should be done for all the variables.

*Both referees commented on this (see RC#24) and we have removed the subjective analyses of excluding 2015 temperature (as there are outliers in other parts of the time-series as well).*

12) Line 231-234. The sentence is quite long and difficult to read, please rephrase it.

*We have edited this text to form two complete sentences.*

13) Line 254. Does "which peaked in June" refer to parameters or to their monthly trends? Section 3.3.

*We revised the sentence to simplify the statement, which now reads, "The fastest increases in $A_T$ and $C_T$ occurred from May through July."*

14) Lines 262 and 274. It is not clear the message that the authors want to convey. At line 262 it is said that salinity is a poor predictor of AT, however the section ends with a salinity-alkalinity regression. Please, review this section consistently.

*We have edited the text to explain (1) that salinity was a poor predictor of $A_T$ over an annual observation period and (2) why we used monthly means to describe the salinity-$A_T$ relationship. For the latter, we now write: "To capture this seasonality without the inter-annual variation of $A_T$, the salinity-$A_T$ relationship at Point B was estimated from monthly means…"*

15) Lines 294-295. The sentence is long and difficult to read. Please rephrase it. Further, what do the authors mean for "morning sampling"? Is it referred to the sampling procedure of the Point B timeseries? If so, it should be introduced in Material & Method, and motivation explained if important.

*We removed this sentence as it does not add important information and this sampling procedure is already described in the Materials and Methods.*

16) Line 311-312. This sentence seems inaccurate. Which is the causal factor of CT increase due to AT increase?

*We have revised this discussion point to clarify that increases in $C_T$ could just be a direct consequence of increasing $A_T$, which has carbon constituents, and include a discussion on various processes that could give rise to these trends. These edits are part of the revised Discussion (2nd, 3rd, and 4th paragraph of the Discussion). We use this relationship between $A_T$ and $C_T$ to make an assumption that allows us to estimate the role of increasing atmospheric $CO_2$ on the ocean acidification trend at Point B.*

17) Line 313. Which "spatial extent" do the authors mean?

*We have removed this sentence for clarity. In the revised Discussion, we simply report other observed trends nearest to Point B (DYFAMED, Adriatic Sea).*

18) Lines 336-337. The analysis of the coastal–offshore gradient would deserve some addition investigations, since offshore deep water is supposed to play a role for CT evolution at point B (at lines 323-325).

*This may be a misunderstanding as we do not intend to indicate influence of offshore deep water at Point B. We have revised the statement about winter mixing offshore to show that deep water has higher $C_T$ and specify that this is observed at DYFAMED: "For $C_T$, peak values occur in winter when the water column is fully mixed. For reference, at DYFAMED, mixing occurs down to more than 2000 m depth and $C_T$ is up to 100 μmol kg$^{-1}$ higher in deep (Copin-Montégut and Bégovic, 2002)."*

19) Lines 359-369. This part could be moved to introduction.

*We have now integrated the text describing coastal ocean acidification trends into the Introduction.*

20) Lines 772-776. Table 3 caption reports not only the description of the table but also comments on results. Please remove the no necessary text.

*These statements in Table 3 no longer apply due to the new deconvolution analyses. Nonetheless, we have taken care to remove statements describing results from Table legends.*

21) Lines 799-801. Figure 6. Please use the caption to describe the plots without describing the results.

*We have removed any potential result statement from the Figure 6 caption.*

**Anonymous Referee #2,**

The article presents, analyses, and discusses the time series of physical parameters and carbonate system properties gathered in a coastal station of the NW Mediterranean Sea and spanning nearly one decade. The analyses on different time scales of ocean acidification, responsible for changes in the marine CO2 system with effects on the dissolved inorganic carbon (CT), partial pressure of CO2 (pCO2), pHT, total alkalinity (AT) and calcium carbonate saturation states is valuable. Actually, sustained observations of inorganic carbon parameters by means of long term time series count on a few sites over the global oceans, and on even fewer in the Mediterranean basin, especially in the coastal area, although such shallow zones can be exposed to intense land sea interactions and to a great complexity of physical and biological processes interacting with ocean acidification. Ocean acidification in coastal zones is remarkably difficult to predict. The present paper can contribute to a better knowledge of the coastal systems vulnerability to ocean acidification, by investigating on the multiple drivers eventually working in this environment. In particular, the analysis of the time series trends (§3.1) appears robust and the decomposition of ocean acidification into the principal drivers is appropriate and informative.

*No response needed.*

The method applied for the deconvolution of pHT and pCO2, (proposed in §3.2 and 3.3,) is new for both the Med Sea and coastal regions, and provides useful indications on different processes driving ocean acidification in this site in comparison with open ocean.

*No response needed.*

The discussion of CT and AT increases as the main drivers of pH decrease (§4.1 and 4.2 ) is well conducted. The proposed attribution of these two concomitant increases (AT and CT) to terrestrial inputs with changing and increasing AT and CT (riverine water and/or to groundwater springs) seems reasonable. The hypothesis is deeply discussed, although not enough supported by the correlation with low salinity (see major comments). Finally, conclusions well enhance the role of land sea interactions. In conclusion, the publication of this article in OS is worth after a few revisions (reported below).
*No response needed.*

Major Comments:

22) Discussion on drivers of AT and CT trends (line 402-439). I agree with the suggestion of Referee #1. The description (in terms of carbonate chemistry) of the river and underground sources in the region of the Bay of Villefranche should be added for a more complete discussion.
*Unfortunately, these types of data (beyond what we included in the original manuscript) are not available for our study site. We added a statement to the Methods to include the distance of the three nearest rivers in the Bay itself (4, 10, and 25 km) and a statement on the lack of data to pursue hypothesis of $A_T$ and $C_T$ trends: "Signatures of limestone erosion can be observed in $A_T$ of nearby rivers (Var, Paillon, and Roya) but detailed time-series are not available. Likewise, riverine influence at Point B has not been quantified."*

23) Then I'm wondering whether results (§3.3) and discussion (§4.2) might benefit from the recent paper by Fry et al. (2015). In the paper authors calculate "Alk1", the same as normalized AT, to remove the contribution of evaporation and precipitation, and calculate "Alk2" (accounting for river AT) to remove the riverine input. This was done because Friis et al. (2003) found that misleading results are produced if normalized alkalinity is used in ocean regions receiving river outflows. . .. It seems to me that you have all the necessary data (Alkm= measured alkalinity, salinity and Alkr=river alkalinity) requested to calculate Alk2. Following the subtraction of these major processes that affects alkalinity at Point B, I would expect: salinity- AT relationship (through the 9 years of time series) improves the positive AT anomaly disappears (or at least decreases).
*The approach of Fry et al. 2015 would reveal insight to the $A_T$ trends at Point B, however, we do not have the necessary data to do this (see RC#22). The river $A_T$ mentioned in the Methods (see RC#9) is based on data that is collected once or twice a year and do not cover the full range of our study period. We agree with both referees that the alkalinity trends could be investigated with the trends in freshwater sources, as is the most likely driver suggested by us as well, but we just do not have the time-series data to do this for this region.*

Minor comments:

24) Lines 222-224. Temperature anomaly increased but this significance was lost with the exclusion of the year 2015. . .. The sentence is not clear to me. Do you mean that excluding the year 2015 (it was exceptionally warm and SST raised to the highest values during summer) there was no increase in temperature anomaly? and therefore the temperature increase cannot be the

driver of observed changes in the carbonate system properties ? In any case please rephrase the sentence.

*While 2015 was an exceptionally warm year, outliers (e.g., warm events) were also present at the beginning of the time-series. Since we did not take a systematic approach to removing outliers, we have removed this secondary analysis and statements from the paper (RC#11). In addition, we have expanded the discussion on the warming trend that occurred over the study period. Following the revised deconvolution analysis (RC#2), temperature is a significant driver of pH trends.*

25) Lines 320-322. During the transition of these processes, salinity decreases to a minimum in May, reflecting the freshwater input that dilutes AT to minimum values. . . Fresh water is able to dilute AT to minimum values if discharging rivers have lower AT than seawater but this might be not true in case of rivers draining carbonatic watershed (later, line 416, authors report that outflowing rivers into the Bay of Villefranche have high AT). This can be misleading, please modify.

*We have clarified that, locally, rivers nearest to Point B have an $A_T$ that is lower than seawater $A_T$. In light of RC#9, we have clarified our definition of 'high' $A_T$ in the following excerpt from the Methods: "Both of these hydrodynamics movements have signatures of river discharge, which for the Mediterranean Sea in general are high in $A_T$ (Copin-Montégut, 1993; Schneider et al., 2007). Limestone erosion can be observed in the $A_T$ of rivers nearest to Point B (Paillon, due 4 km West; Var due 10 km West; and Roya due 26 km East). River $A_T$ ranges between 1000 to 2000 µmol kg$^{-1}$ (data from Agence de l'Eau Rhône-Méditerranée-Corse, http://sierm.eaurmc.fr), and is slightly lower than seawater $A_T$."*

26) Line 325-326. Following winter CT declines due to a combination of phytoplankton bloom carbon uptake and freshwater dilution. . . Again I would be more cautious as freshwater dilutes if it contains lower CT . . .

*As the $A_T$ of local rivers is ~1000-2000 umol kg$^{-1}$, $C_T$ in river water is not likely to be greater than that of seawater. We have added the assumption that river $C_T$ is lower than seawater $C_T$ to this sentence. "Following winter, $C_T$ declines due to a combination of phytoplankton bloom carbon uptake and freshwater dilution (assuming river $C_T$ < seawater $C_T$), until the onset of summer stratification."*

27) Lines 351-357. The correlates between Point B and N Adriatic Sea suggest a common driver of changes in ocean carbonate chemistry at these two sites (possibly linked via shared watersheds of the Alps) . . . . . ... Mediterranean Sea. Could you explain and present more clearly this hypothesis?

*In response to RC#6, we have removed the qualitative statements comparing Point B with the Northern Adriatic Sea. Instead, we suggest that the two independent studies might indicate a process that is present around a greater coastal region of the Mediterranean Sea.*

*References:*

*Bates, N. R., Astor, Y. M., Church, M. J., Currie, K., Dore, J. E., González-Dávila, M., Lorenzoni, L., Muller-Karger, F., Olafsson, J., and Santana-Casiano, J. M.: A time-series*

view of changing ocean chemistry due to ocean uptake of anthropogenic CO2 and ocean acidification, Oceanography, 27, 126-141, 2014.

Copin-Montégut, C.: Alkalinity and carbon budgets in the Mediterranean Sea, Global Biogeochem. Cycles, 7, 915-925, 10.1029/93GB01826, 1993.

Kapsenberg, L., and Hofmann, G. E.: Ocean pH time-series and drivers of variability along the northern Channel Islands, California, USA, Limnol. Oceanogr., 61, 953-968, 10.1002/lno.10264, 2016.

Schneider, A., Wallace, D. W. R., and Körtzinger, A.: Alkalinity of the Mediterranean Sea, Geophys. Res. Lett., 34, L15608, 10.1029/2006GL028842, 2007.

---

## Author Response (AR2)

**Author Response to referee comments on "Coastal ocean acidification and increasing total alkalinity in the NW Mediterranean Sea" by Kapsenberg et al.**

**Overall author response:** We thank the referee for his/her time and the detailed oriented suggestions which have improved the manuscript. For the ease of reviewing our response, we have numbered the Referee Comments from 1 to 17 and refer to these numbers in the Tracked Changes revised document and in our point-by-point response below in blue.

**The major revision** includes a revised calculation of the deconvolutions according to Garcia-Ibanez et al. (2016), which are now displayed in separate table for the pH and pCO2 deconvolution (Table 3 and 4). As expected, the revised calculations that closely match those of Garcia-Ibanez produced the same results as our original approach, which made an estimate of the partial derivatives without calculating the sensitivity component first. As such, no substantial changes were needed in the Results or Discussion.

**Anonymous Referee #1 Submitted on 27 Feb 2017**

The authors did a valuable work to clarify several aspects of their analysis and to improve the readability of the manuscript. Further, they provided detailed and convincing answers to the reviewers' comments supporting their thoughts. However, a couple of points (point 2 and 4 of their response letter) of the previous review still need to be addressed accurately and are source of concerns before the manuscript can be accepted for publication.

**Regarding point 2 of the response letter.**

**1**) L10P196-215: the description of the deconvolution needs further explanation. The answer provided by the authors is not satisfactory.

To my understanding of this section, the authors computed 4 timeseries of pH by using: the true observations of T and the climatological monthly means of the other three variables; the true observations of S and the climatological monthly means of the other three variables, and so on for DIC and ALK.

Then, the linear regression analysis is performed on these 4 pH timeseries, and the estimated slopes are reported in Table 3. Therefore, I think that this is not the methodology proposed by Garzia-Ibanes et al., (2016). Indeed, the sensitivity of the pH to the four drivers (i.e. the changes of pH caused by the changes in the drivers () as prescribed in Garzia-Ibanes' method) is not provided in the manuscript. Is it perhaps computed? Then, the different pH contributions are not computed as the multiplication of times the trend of the variables. Therefore, the equation (1) is not correct with respect to the analysis that is effectively performed, and some of the results and conclusions should be provided in a different way.

I suggest that the analysis presented in the manuscripts can be maintained, however authors should provide the robustness of the underpinning hypothesis (i.e. the overall trend can be decomposed in such a way) and they should provide the sensitivity of the method to the choice of the temporal means of the variables that are kept constant (i.e. how different would be the results if the averages are computed on yearly or over the whole period?). Revise accordingly this section and the results.

Our original approach estimated each partial derivative in one step by performing one regression. We have now closely followed the method of Garcia-Ibanez et al. (2016) by solving for each component of each partial derivative using the observed values of one variable and mean values (all samples from 2009-2015) for the remaining three variables.

The new calculation essentially produced the same results. Small numerical differences have been updated throughout the Abstract, Results, and Discussion. The agreement between both analytical methods was expected given that the method of Garcia-Ibanez et al. (2016) was the mathematical underpinning of our original approach.

P13L272-278: provided that the deconvolution is computed as previously described, these comments should be revised since these numbers are not produced by the multiplication of times (i.e. the trends of T, S, alkalinity and DIC reported in table 2 and the sensitivity of pH to the vars).

Since we have updated the analysis, this is no longer relevant.

L271: why do these results indicate that the deconvolution analyses well represent the observed trend?

*We now specify that the "deconvolution reproduced influences of temperature sensitivity well".*

**Regarding point 4 of the response letter.**

**2**) At P13L282-284 it is said that the trend of atmospheric CO2 represents the "maximum influence of anthropogenic CO2 forcing at Point B" under the assumption that "increase in atmospheric CO2 causes an equal increase in seawater pCO2". This assumption is quite questionable. The authors should provide some evidences in support of this assumption.

We have removed the word "maximum" as the emphasis is on the fact that the waters at 1 m are likely in equilibrium with the atmosphere on annual time-frames. We have added a reference showing evidence for this.

Then, at P16L345-347, it is argued that the atmospheric pCO2 increase is the remaining part composing the contribution of  $\Delta$ CT to  $\Delta$ pCO2, which assumes that the atmospheric  $\Delta$ pCO2 is the actual contribution and not, as previously hypothesized, the maximum one. As the authors surely understand, the similarity between two numbers does not imply any physical relationship. Therefore, this conclusion seems not supported by the results. Please resolve it.

The matching of these rates, Henry's Law, and the trends of surface samples all provide strong evidence at the study site and at annual time-frames, surface water is in CO2 equilibrium with the atmosphere. This was shown previously using the first four years of this time-series (De Carlo et al. 2013). We have not changed our conclusion but we have now added the De Carlo et al. (2013) reference and included a statement indicating that the closeness of the rates does not imply causation.

Further, at L347-349: which is the causal relationship between the influence of atmospheric pCO2 and the significance of monthly CT trends? The increase of atmospheric pCO2 should have an effect throughout the year (in winter and autumn months too). Therefore, results do not show the influence of atmospheric pCO2 to the significance of monthly CT trends.

L347-349 read, "The influence of atmospheric CO2 can also be observed in the significance of monthly CT trends (eight months out of the year) compared to AT trends (three months out of the year, Fig. 5)."

The point here is that the increase in CT is distributed over a longer period throughout the year than the increase in AT. The added input of atmospheric CO2 potentially pushed monthly CT increase into a statistically significant category. We provide quantitative evidence for this idea using the significant increases shown in Fig. 5. We have clarified this approach with the revised sentence: "Monthly  $C_T$  trends are positive and statistically significant over more months than  $A_T$  trends (8 vs. 3 months), which are more seasonally restricted (Fig. 5)."

Finally, L354-356: since it has not been demonstrated that the subtraction of delta AT from delta CT gives the contribution of atmospheric pCO2 (i.e. trends in other processes can have contributed), these sentences appear poorly supported by the results.

Given what is known of ocean chemistry and all other parameters involved in the carbonate system, biology, and oceanography at Point B, we have put all the proper caution and outlined all assumptions surrounding this approach for calculations at 1 m. In addition, the referee has not provided any evidence of alternative explanations that would refute our conclusions. We have already stated the assumptions of this method for data from 1 m depth, and again emphasize that this is a "simple model" when presenting the Discussion on data from 50 m.

**Other minor points**

**3) Abstract**

L2P27-28: CT increase could be driven by the same processes that caused the increase in At not by the increase of At itself.

We have edited the sentence to clarify this, "The remaining  $C_T$  increase may have been driven by the same unidentified process that caused an increase in  $A_T$ ".

**4)** L2P34-35: the conclusion about rapid warming could be misleading by the fact that the length of the timeseries is very short. As well, also some of the trend values reported in Table 2 appear quite large. I wonder whether the length of the timeseries (only 9 years) could have played any role in overestimating the slopes, since it seems (after a simple visual inspection of Fig. 2) that some timeseries have a maximum in 2012 -2013 and values do not increase more after that period. *AT and CT have high values in 2012-2013, but these are not independent from one another (i.e. these parameters have shared ion constituents) and the remaining parameters do not show particularly unusual patters in 2012-2013. In addition, we have discussed how the process influencing AT and CT trends might be wholly independent from the other trends. I would suggest testing the robustness of the trends by using a bootstrap analysis (or any other re-sampling technique) or testing a regime shift*

analysis to verify whether it is a trend or a regime shift.

I acknowledge that authors specify that trends (i.e. warming, and acidification) referred to the specific 2007-2015 period in several parts of the manuscript, however, a comment about the reliability of trends computed on very short timeseries should be added somewhere in the manuscript (e.g. at L363-365 and in the conclusion).

The data show that rapid warming occurred over the study period, whether or not the study period is considered short or long. It is unclear which specific trends the referee is referring to in the comments that trends "appear quite large".

Only time will tell how consistent these trends are over time. We are simply describing changes that have occurred over the study period (the longest for the coastal Mediterranean Sea).

**5)** P8L140-141: avoid to use the "river signature" of the Mediterranean Sea while the Bay of Villefranche is described.

We have removed this statement from the Methods.

**6) P8L144: provide the position of rivers in Figure 1**

We did not include a map of local rivers because: (1) we do not present time-series data from rivers, (2) we do not have data that indicate which are the influential rivers (which could also be larger rivers further away), and (3) we only discuss rivers as a potential influence on the observed trends. For this reason, distance and direction of local rivers nearest to the study sites was already reported in Section 2.1, "Paillon, due 4 km West; Var due 10 km West; and Roya due 26 km East".

7) P10L190: Should it be called "climatological monthly means"? Even if the word "climatological" is referred to a temporal average over longer periods than the presently considered 9-year period, the use of "monthly means" can be misleading. Provide a definition of how anomalies are computed (i.e., at L190, L212 and L215).

L190: We have revised the text in the Methods (Section 2.2) to read: "To quantify interannual changes in carbonate parameters, the data were detrended for seasonality by subtracting the respective climatological monthly means computed for the period 2009-2015 from the time-series ('monthly means' from hereon)."

L212: We have now written: "For these data, missing daily values were linearly interpolated, monthly means were calculated and subtract from the time-series to generate an anomaly time-series." We do not specify climatological monthly means as this definition is defined previously.

*L215:* We have clarified: "...linear regressions were performed on changes in  $A_T$  and  $C_T$  by month (mean value of observations within one month) from 2009 through 2015..."

8) P12L243: more than 400 samples. *Done*.

**9)** P12L244: do the authors mean that the trends of all variables are significant both at 1 and 50 m and only Salinity at 1 m is not significant? The sentence is not very clear. Then, avoid writing all trends estimates (and confidence interval and number of points) since they are already shown in Table 2. Provide the number of points in Table 2 along with the unit of the variable trends.

The referee has correctly interpreted this sentence. We have simplified the reporting of trends in paragraph form. We have now added the N values to Table 2. Unit variables have been moved from the Table 2 legend into the table.

**10)** Table 2: Which is the meaning of "Total" in the first column? are you meaning the whole Mediterranean Sea?

We assume the referee is referring to Table 1 for this comment. Total does indeed mean the whole Mediterranean Sea. We have added this clarification to the Table 1 legend.

**11**) P15L311: provide an appropriate symbol for AT and S that indicates that AT and S are monthly means. Then, more importantly, add a new plot to figure 4 reporting the regression between salinity and alkalinity based on the monthly means.

We have revised the alkalinity and salinity variables reported in Equation 2 to identify that these are monthly mean values. This equation still has a wide margin of error and given the importance of interannual variability at this site as well, it is not a focal point of our manuscript.

**12**) P15L328-329 and Fig. S1: it would be interesting to see some statistical tests on the relationship between diel pHT variability and T and Chl variability. Or just do not mention it. The Figure S1 does not show a clear message.

We have analyzed the data with a linear regression but these are not statistically significant. We have removed the figure from the supplemental text and only state that diel pH variation is unrelated to temperature and Chl-a (data not shown).

**13**) I acknowledge that the authors have chosen to not further investigate the high frequency pH time-series, however, I suggest adding at least the estimate of the diel variability of pH to be compared with the trend estimate (L268) and the annual range (L263). Providing the different scales of variability of pH (daily, seasonal, interannual) would be of great interest for many readers.

We already both reported (Section 3.4) and plotted (Figure 7) diel pH variability for this site. But to put all numbers together, we have added a sentence summarizing pH variations over different timescales in the first paragraph of the Discussion and a short statement in the Abstract.

**14)** L338-341: the assumption that the increase in AT is due to increases in its carbon constituents deserve a better verification. Since HCO3- and CO32- are computed by SeaCarb, the no-carbonate AT can be easily derived and the regression of no-carbonate AT can be calculated in order to verify the assumption. Otherwise, some of the conclusions that follow (e.g. L350-353) cannot be validated by the results.

Computing the non-carbonate  $A_T$  does not reveal an increase in  $A_T$  (see figure below). This is not surprising as what is left is the alkalinity of borate (a function of salinity which changes very little at our study site) and the contribution of silicate and phosphate which are extremely small at an oligotrophic site such as Point B. We believe that this result is trivial and does not warrant a mention in the manuscript.

**15)** L366-370: which is the rationale of the relationship between the temperature increase and climate indexes? Provide any statistical correlation index.

Using the same reference, we have clarified that both the AMO and NAO are associated with episodic warming of the Mediterranean Sea.

**16)** P18L391 and L410-411: provide a short definition of "atmospheric forcing". Do the authors refer to CO2 exchange, Evaporation minus Precipitation, and warming? This definition should be introduced in the abstract at P2L26.

We have now specified CO2 forcing in all instances.

**17**) Plots b, d and e of Figure 7 are never introduced nor commented in the text. Remove them if not needed.

The content of Figure 7 was discussed throughout Section 3.4, but we have now explicitly added "Fig. 7x" throughout this portion of the text.

- 1 Coastal ocean acidification and increasing total alkalinity in the NW Mediterranean Sea
- 2
- 3 Lydia Kapsenberg1, Samir Alliouane1, Frédéric Gazeau1, Laure Mousseau1, and Jean-Pierre
- 4 Gattuso1,2,§
- 5
- 6 1Sorbonne Universités, Université Pierre et Marie Curie-Paris 6, CNRS-INSU, Laboratoire
- 7 d'Océanographie de Villefranche, 06230, Villefranche-sur-Mer, France
- 8 2Institute for Sustainable Development and International Relations, Sciences Po, 27 rue Saint

[revised manuscript text omitted]

- 177 Service National d'Analyse des Paramètres Océaniques du CO2, at the Université Pierre et
- 178 Marie Curie in Paris, France. Precision of  $C_{\rm T}$  and  $A_{\rm T}$  was less than 3 µmol kg-1, and the

Comment [LK3]: <mark>#5</mark>

Comment [LK4]: Dates: 2014 03 24, and 2013 05 20

| 181 | average accuracy was 2.6 and 3 $\mu$ mol kg -1 , as compared with seawater certified reference                                      |
|-----|------------------------------------------------------------------------------------------------------------------------------------------------|
| 182 | material (CRM) provided by A. Dickson (Scripps Institution of Oceanography). Only                                                              |
| 183 | obvious outliers were omitted from the analyses: three $C_{\rm T}$ values at 1 m (> 2300 $\mu$ mol kg -1 ),                         |
| 184 | one $A_{\rm T}$ value at 1 m (> 2900 $\mu$ mol kg -1 ), and one $A_{\rm T}$ value at 50 m (< 2500 $\mu$ mol kg -1 ). The |
| 185 | $C_{\rm T}$ and $A_{\rm T}$ measurements on replicate bottle samples were averaged for analyses.                                               |
| 186 | Calculations of the carbonate system parameters were performed using the R package                                                             |
| 187 | seacarb version 3.1 with $C_T$ , $A_T$ , in situ temperature and salinity as inputs (Gattuso et al.,                                    |
| 188 | 2016). Total concentrations of silicate (SiOH 4 ) and phosphate ( $PO_4^{3-}$ ) were used when                                      |
| 189 | available from Point B (L. Mousseau, unpubl., http://somlit.epoc.u-bordeaux1.fr/fr/).                                                          |
| 190 | Detection limits for nutrients were 0.03 $\mu$ M for SiOH 4 and 0.003 to 0.02 $\mu$ M for PO 4 3- ;           |
| 191 | relative precision of these analyses is 5-10 % (Aminot and Kérouel, 2007). Total boron                                                         |
| 192 | concentration was calculated from salinity using the global ratio determined by Lee et al.                                                     |
| 193 | (2010). The following constants were used: $K_1$ and $K_2$ from Lucker et al. (2000), $K_f$ from                                               |
| 194 | Perez and Fraga (1987), and $K_s$ from Dickson (1990). Reported measured parameters are                                                        |
| 195 | temperature, salinity, $A_T$ , and $C_T$ , and derived parameters are pH T (total hydrogen ion scale),                              |
| 196 | $pH_T$ normalized to 25 °C (pH_{T25}), pCO_2, and aragonite ( $\Omega_a)$ and calcite ( $\Omega_c)$ saturation states.                         |
| 197 | Salinity-normalized changes in $A_T$ (n $A_T$ ) and $C_T$ (n $C_T$ ) were calculated by dividing by in situ                             |
| 198 | salinity and multiplying by 38. Except for $pH_{T25}$ , all parameters are reported at in situ                                          |
| 199 | temperatures.                                                                                                                                  |
| 200 | The average uncertainties of the derived carbonate parameters were calculated                                                                  |
| 201 | according to the Gaussian method (Dickson and Riley, 1978) implemented in the "errors"                                                         |
| 202 | function of the R package seacarb 3.1 (Gattuso et al., 2016). The uncertainties are $\pm$ 2.7 x 10 -                                |
| 203 | $^{10}$ mol H $^{+}$ (about 0.015 units pH $_{T}), \pm 15$ µatm pCO $_{2},$ and $\pm$ 0.1 unit of the aragonite and calcite                    |

saturation states.

| 205 | To quantify interannual changes in carbonate parameters, the data were detrended for                                                                                                                                            |          | Comment [LK5]: <mark>#7</mark>                                                                    |
|-----|---------------------------------------------------------------------------------------------------------------------------------------------------------------------------------------------------------------------------------|----------|---------------------------------------------------------------------------------------------------|
| 206 | seasonality by subtracting the respective climatological monthly means, computed for the                                                                                                                                        |          | Deleted: s                                                                                        |
| 207 | period 2009-2015 from the time-series ('monthly means' from hereon). The resulting                                                                                                                                              |          |                                                                                                   |
| 208 | residuals, were analyzed using a linear regression to compute anomaly trends. This approach                                                                                                                                     | 1        | Deleted: anomalies                                                                                |
| 209 | follows methods from Bates et al. (2014) to allow for comparisons of trends observed at                                                                                                                                         |          |                                                                                                   |
| 210 | different time-series stations. All analyses were performed in R (R Core Team, 2016).                                                                                                                                           |          |                                                                                                   |
| 211 |                                                                                                                                                                                                                                 |          |                                                                                                   |
| 212 | 2.3. Deconvolution of pH T and pCO 2                                                                                                                                                                      |          | Comment [LK6]: <mark>#1</mark>                                                                    |
| 213 | To identify proportional contributions of various drivers to ocean acidification trends                                                                                                                                         |          |                                                                                                   |
| 214 | at Point B, deconvolution of time-series $pH_T$ and $pCO_2$ was performed following methods                                                                                                                                     |          |                                                                                                   |
| 215 | from García-Ibáñez et al. (2016) for observations at 1 and 50 m. The equation is described                                                                                                                                      |          |                                                                                                   |
| 216 | below for $pH_T$ , where changes in $pH_T$ are driven by changes in temperature ( T ), salinity ( S ),                                                                                                            |          |                                                                                                   |
| 217 | $A_{\rm T}$ , and $C_{\rm T}$ , over time (t), according to the following model:                                                                                                                                                |          |                                                                                                   |
| 218 | $\frac{dpH_T}{dt} = \frac{\partial pH_T}{\partial T}\frac{dT}{dt} + \frac{\partial pH_T}{\partial S}\frac{dS}{dt} + \frac{\partial pH_T}{\partial A_T}\frac{dA_T}{dt} + \frac{\partial pH_T}{\partial C_T}\frac{dC_T}{dt} $ (1) |          |                                                                                                   |
| 219 | Here, $\frac{\partial p H_T}{\partial var} \frac{dvar}{dt}$ represents the slope contribution of changing var to the estimated                                                                                           |          |                                                                                                   |
| 220 | change in pH T ( $\frac{dpH_T}{dt}$ ), where var is either temperature (T), salinity (S), $A_T$ , or $C_T$ . The                                                                                                     |          |                                                                                                   |
| 221 | sensitivity of pH to var $\left(\frac{\partial pH_T}{\partial nar}\right)$ was estimated by calculating pH T using the true observations                                                                      | 1        | Deleted: rate of                                                                                  |
| 222 | of var and holding the other three variables constant (mean value of the time-series) and                                                                                                                                       |          | Deleted: change due
|     |                                                                                                                                                                                                                                 |          | Deleted: but                                                                                      |
| 223 | regressing it to var. Sensitivity $\left(\frac{\partial pH_T}{\partial r}\right)$ was then multiplied by the anomaly rate of var (Table                                                                                         |          | Deleted: monthly mean values                                                                      |
|     |                                                                                                                                                                                                                                 |   | Deleted: of the other three variables                                                             |
| 224 | 2). The calculation was repeated for pCO 2 $\left(\frac{dpCO_2}{dt}\right)$ in order to compare the rate of increase                                                                                          |          | Deleted: , and regressing the anomaly (following subtraction of respective monthly means) to time |
| 225 | with that of atmospheric CO 2 .                                                                                                                                                                                      |          |                                                                                                   |
| 226 | As a sub-component of $\frac{\partial p C O_2}{\partial C_T} \frac{d C_T}{dt}$ , the rate of anthropogenic CO 2 increase was                                                                                         |          |                                                                                                   |
| 227 | estimated from atmospheric CO 2 concentrations nearest to Point B (Plateau Rosa, Italy,                                                                                                                              |          |                                                                                                   |

- 238 courtesy of the World Data Center for Greenhouse Gases,
- 239 http://ds.data.jma.go.jp/gmd/wdcgg/). For these data, missing daily values were linearly
- 240 interpolated, monthly means were calculated and subtracted from the time-series to generate
- 241 an anomaly time-series. A linear regression was performed on anomalies where the slope
- 242 represents the rate of anthropogenic CO2 increase in the atmosphere. Finally, to help identify
- 243 different processes that might have contributed to the observed trends, linear regressions were
- performed on changes in  $A_{\rm T}$  and  $C_{\rm T}$  per month (mean value of observations within one

245 month) from 2009 through 2015 and on the salinity- $A_{\rm T}$  relationship by year.

246

**247 2.4. SeaFET data collection, processing, and analysis**

248 To capture pH variability at higher-than-weekly sampling frequencies, a SeaFET™ 249 Ocean pH sensor (Satlantic) was deployed on the EOL buoy (435 m from the Point B 250 sampling site) starting in June 2014, at 2 m depth. Autonomous sampling was hourly and 251 deployment periods ranged between 1 and 3 months. Field calibration samples for pH were 252 collected weekly, using a Niskin bottle next to SeaFET within 15 min of measurement. This 253 sampling scheme was sufficient for this site as there is no large high-frequency pH 254 variability. Unlike Point B sampling, SeaFET calibration samples were processed for pH 255 using the spectrophotometric method (Dickson et al., 2007) with purified m-cresol purple 256 (purchased from the Byrne lab, University of South Florida). In situ temperature, salinity, and 257 AT measured at Point B, within 30 min of the SeaFET sampling, were used to calculate in situ 258 pHT of the calibration samples. SeaFET voltage was converted to pHT using the respective 259 calibration samples for each deployment period, following the methods and code described in 260 Bresnahan et al. (2014) but adapted for use in R.

261 The estimated standard uncertainty in SeaFET  $pH_T$  is  $\pm 0.01$  and was calculated as the 262 square root of the sum of each error squared. The sources of errors are: measurement error of

|
                           |
|--------------------------------|
| Comment [LK7]: <mark>#7</mark> |
|                                |
|                                |
|                                |
|                                |
| Comment [LK8]: <mark>#7</mark> |
|                                |
|                                |
|                                |

- 264 spectrophotometric pH ( $\pm$  0.004, N = 68 mean SD of 5 replicate measurements per calibration
- sample for samples collected between 16 July 2014 and 3 May 2016), spatio-temporal
- 266 mismatch sampling at EOL ( $\pm$  0.007, mean offset of pHT of the calibration samples from
- 267 calibrated time-series), and variability in purified m-cresol dye batch accuracy as compared
- 268 to Tris buffer CRM pH ( $\pm$  0.006, mean offset of pHT of the spectrophotometric measurement
- 269 of Tris buffer from the CRM value).
- 270
- 271 3. Results
- 272

**273 3.1. Time-series trends**

274 At Point B from January 2007 to December 2015, more than 400 samples were 275 collected for carbonate chemistry at both 1 and 50 m. Anomaly trends detected at 1 m (Fig. 2) 276 were also significant at 50 m (Fig. 3, Table 2), with the exception that salinity increased at 50 277 m (0.0063  $\pm$  0.0020 units yr-1). At 1 m, trends were significant for pHT (-0.0028 units yr-1),  $A_{\rm T}$  (2.08 µmol kg-1 yr-1),  $C_{\rm T}$  (2.97 µmol kg-1 yr-1), pCO2 (3.53 µatm yr-1), and  $\Omega_{\rm a}$  (-0.0064 278 279 units yr-1). At the same time, temperature anomaly increased (0.072 °C yr-1), but no 280significant change in the salinity was detected at 1 m. Trends of carbonate chemistry parameters were faster at 1 m compared to 50 m, with the exception of salinity and 281 282 temperature. The warming rate at 50 m was slightly greater compared to 1 m, mostly due to 283 increasing summer temperatures since 2007. 284 Strong seasonal cycles of carbonate chemistry parameters were present at Point B at 1 285 m (Fig. 4). Climatological monthly means (2007-2015) are described briefly and listed in Table S1. Mean temperature range was 11.2 °C with a maximum at 24.77  $\pm$  1.35 °C in August 286 287 and minimum of  $13.58 \pm 0.41$  °C in February. The range in  $A_{\rm T}$  was 19 µmol kg-1 from June to September. The  $C_{\rm T}$  range was 33 µmol kg-1 with a peak in late winter and minimum values in 288

Comment [LK9]: <mark>#</mark> Deleted: >

| Deleted: , $P = 0.002$          |
|----------------------------------------|
| Deleted: ± 0.0003               |
| Deleted: , $N = 412$            |
| Deleted: ± 0.19                 |
| Deleted: , $N = 417$            |
| Deleted: $\pm 0.20$             |
| Deleted: , $N = 416$            |
| Deleted: ± 0.39                 |
| Deleted: , $N = 412$            |
| Deleted: $\pm 0.0015$           |
| Deleted: , $N = 412$            |
| Deleted: ± 0.022                |
| Deleted: , $N = 413$            |
| Deleted: and                           |
| Deleted: $(P = 0.702, N = 417)$ |
| Deleted: Calculated                    |

| 306 | August and October. Due to summer warming coinciding with the period of peak primary                                              |                                                                                                                                                                                                                 |
|-----|-----------------------------------------------------------------------------------------------------------------------------------|-----------------------------------------------------------------------------------------------------------------------------------------------------------------------------------------------------------------|
| 307 | productivity (De Carlo et al. 2013), warming countered the influence of low $C_T$ on pH. As a                                     |                                                                                                                                                                                                                 |
| 308 | result, $pH_{\underline{T}}$ reached minimum values in summer (8.02 ± 0.03, July and August) and peaked in                        |                                                                                                                                                                                                                 |
| 309 | late winter (8.14 $\pm$ 0.01, February and March), for an overall annual pH range of 0.12. The                                    |                                                                                                                                                                                                                 |
| 310 | corresponding pCO2 range was 128 µatm from February to August. Seasonal cycles were                                               |                                                                                                                                                                                                                 |
| 311 | smaller at 50 m compared to 1 m (Table S1).                                                                                       |                                                                                                                                                                                                                 |
| 312 |                                                                                                                                   |                                                                                                                                                                                                                 |
| 313 | 3.2 Deconvolution of pHz and pCO                                                                                                  |
Comment II K101: #1                                                                                                                                                                                         |
| 515 |                                                                                                                                   |                                                                                                                                                                                                                 |
| 314 | Deconvolutions of pH and $pCO_2$ are presented in Table 3 and 4, respectively. The                                                |                                                                                                                                                                                                                 |
| 315 | estimated anomaly trends $\left(\frac{dpH_T}{dt}, \frac{dpCO_2}{dt}\right)$ from the deconvolution fall within the error of the   |
| 316 | observed anomaly trends (Table 2). The contribution of warming to the pH T anomaly ( $\zeta$                           |
| 317 | 0.0011 units $yr^{-1}$ , at 1 m) matched the difference between the trends of pH T and pH T25C (Table       |
| 318 | 2), which verifies that the deconvolution reproduced influences of temperature sensitivity                                        |                                                                                                                                                                                                                 |
| 319 | well. Overall, these results indicate that the deconvolution analyses represent the observed                                      |
| 320 | trends well.                                                                                                                      |                                                                                                                                                                                                                 |
| 321 | At both depths, the predominant driver of $\frac{dpH_T}{dt}$ and $\frac{dpCO_2}{dt}$ was the increase in $C_T$ .                  |
| 322 | Increasing $A_T$ countered 66-69 and 60 % of the influence of increasing $C_T$ on $\frac{dpH_T}{dp}$ and $\frac{dpCO_2}{dp}$      | $\frac{dt}{dt}$ and $\frac{dt}{dt}$ , respectively ( $P = 0.001$ ). Since warming was slightly greater at 50 m compared to 1 m, warming accounted for a larger proportional influence on $\frac{dpH_T}{dt}$ and |
|     | $\frac{dt}{dt} = \frac{dt}{dt}$                                                                                                   | $\frac{dpCO_2}{dt}$ at 50 m compared to 1 m.                                                                                                                                                                    |
| 323 | respectively. At 1 m, warming accounted for $\frac{41}{2}$ and 37 % of $\frac{dpH_T}{dt}$ and $\frac{dpCO_2}{dt}$ , respectively. | Comment [LK11]: = 100* [%contribution of
At]/[%contribution Ct]                                                                                                                                              |
| 324 | Since warming was slightly greater at 50 m compared to 1 m, warming accounted for a larger                                        | for both pH and pCO2 deconv. Deleted: by 64 and 60 %,                                                                                                                                                           |
| 325 | proportional influence on $\frac{dpH_T}{dt}$ and $\frac{dpCO_2}{dt}$ at 50 m compared to 1 m. Increasing salinity at 50           | Deleted: at 1 m and by 65 and 58 %, respectively at 50 m
|     | ut ut                                                                                                                             |                                                                                                                                                                                                                 |
| 326 | m contributed slightly to $\frac{dpH_T}{\sqrt{dt}}$ (4 %) and $\frac{dpCO_2}{dt}$ (2 %).                                          |
| 327 | Atmospheric CO 2 anomaly at Plateau Rosa increased by $2.08 \pm 0.01$ ppm yr -1 ( $F_{1,3285}$              |                                                                                                                                                                                                                 |
| 328 | = 4664, $P \ll 0.001$ , $R^2 0.93$ ) during the study period 2007-2015, and represents the                                        |                                                                                                                                                                                                                 |
| 329 | anthropogenic CO 2 forcing on seawater pH. To estimate the influence of anthropogenic CO 2                  |

- 344 forcing at Point B, we assume air-sea CO2 equilibrium (e.g., increase in atmospheric CO2
- causes an equal increase in seawater pCO2) for the water mass at 1 m. This assumption is
- based on evidence that Point B is a weak sink for atmospheric CO2 with near-balanced air-sea
- 347 CO2 flux on an annual time-frame (De Carlo et al., 2013). Considering the error associated
- 348 with deconvolution of pCO2 at 1 m, atmospheric CO2 increase can, at most, represent 38-43
- 349 % of the total  $C_{\rm T}$  contribution  $\left(\frac{\partial p C O_2}{\partial C_T} \frac{d C_T}{dt}\right)$  to  $\frac{d p C O_2}{dt}$ . This leaves 57-62 % of the total  $C_{\rm T}$
- 350 contribution to  $pCO_2$  trends unaccounted for.
- 351 As  $A_{\rm T}$  is not influenced by addition of anthropogenic CO2 to seawater but still
- 352 increased, the next question was whether or not the changes in  $A_{\rm T}$  and  $C_{\rm T}$  were process-
- 353 linked. At 1 m, regressions of annual monthly trends of  $A_{\rm T}$  and  $C_{\rm T}$  revealed similar seasonal
- 354 cycles for both parameters (Fig. 5, Table S2). The fastest increases in  $A_{\rm T}$  and  $C_{\rm T}$  occurred
- 355 from May through July. The smallest (non-significant) changes occurred in January. The
- 356 synchronicity between monthly trends of  $A_{\rm T}$  and  $C_{\rm T}$  was also observed at 50 m, but the rates 357 were slower (analysis not shown).
- 358

**359 **3.3. Salinity and** *A*T **relationships**

| 360 | Over an annual observation period at 1 m, salinity was a poor predictor of $A_{\rm T}$ , with the      |
|-----|--------------------------------------------------------------------------------------------------------|
| 361 | exception of 2007 (Fig. 6). The $R^2$ value for each annual salinity- $A_T$ relationship at 1 m        |
| 362 | ranged from 0.00 (in 2013) to 0.87 (in 2007) with y-intercepts ( $A_{T0}$ , total alkalinity of the    |
| 363 | freshwater end-member) ranging between -176 $\mu mol~kg^{-1}$ (in 2007) and 2586 $\mu mol~kg^{-1}$ (in |
| 364 | 2013). The interannual variability of the salinity- $A_T$ relationship was driven by the variability   |
| 365 | in $A_{\rm T}$ observed at salinity < 38.0 that was present from November through July.                |
| 366 | Changes in salinity (based on monthly means) at Point B was small and ranged from                      |
| 367 | $37.64\pm0.26$ to $38.21\pm0.11$ from May to September, following freshwater input in winter           |
|     |                                                                                                        |

and spring and evaporation throughout summer and fall (Fig. 4). Highest (> 38.0) and most

Comment [LK13]: #

| {     | Deleted: 7                                                        |
|-------|-------------------------------------------------------------------|
| ····{ | Deleted: 2                                                        |
| -     | Comment [LK14]: =2.08/(5.14 + 0.35)
=2.08/(5.14 - 0.35) |
| T     | Deleted: 8                                                        |
| Ì     | Deleted: 3                                                        |

[revised manuscript text omitted]

positive since the 1990s1 and positive NAO phases were prevalent during the second half of
our study2.

[revised manuscript text omitted]

---

## Author Response (AR3)

**Author Response to referee comments on "Coastal ocean acidification and increasing total alkalinity in the NW Mediterranean Sea" by Kapsenberg et al.**

*Overall author response: We thank the Editor for his time and the detailed comments for corrections. For the ease of reviewing our response, we have numbered comments from 1 to 18 and refer to these numbers in the Tracked Changes revised document and in our point-by-point response below in blue.*

1. L90 these studies cover various periods … (delete: study ; twice study in this sentence is not necessary). *Deleted "study".*

2. L146-148 "The site Point B is an historical sampling point, since 1957, regarding several oceanographic parameters." Is this sentence ok like this? Looks strange to me. *Revised sentence is: "Point B has been an oceanographic station since 1957".*

3. L150 "predominantly" is definitely more common. *The correction has been made.*

4. L178-182 "Precision of CT and AT was less than 3 μmol kg-1, and the average accuracy was 2.6 and 3 μmol kg-1, as compared with seawater certified reference material (CRM) provided by A. Dickson" To my understanding, the accuracy must always be less good (i.e., higher absolute number) than the precision, because it is superimposed on the precision. *Accuracy was determined as the mean offset between the measured and reference value of the CRMs. Precision refers to the repeatability of attaining the same value across replicate bottle samples (which includes errors associated with sample collection, preservation, etc.). We have updated the text to use the phrase repeatability instead of precision: "Average accuracy of $C_T$ and $A_T$ measurements was 2.6 and 3 μmol $kg^{-1}$, respectively, as compared with seawater certified reference material (CRM) provided by A. Dickson (Scripps Institution of Oceanography). Repeatability of replicate samples was better than 3 μmol $kg^{-1}$."*

5. L276 … that salinity only increased … (suggest to add: only). *Done.*

6. L280 trend (instead of: change)? *Done.*

7. L281 greater (instead of: faster)? *Done.*

8. L351-352 … but still it increased … (add: it)? *Replaced with "but it did increase"*

9. L366 were (not: was) *Done.*

10. L385-391 This part must be moved to the Materials and Methods section. *The information in this section has been moved to the Methods section.*

11. L447-448 … which may result from warming due to increased evaporation) (I think it is useful to add this is not just warming but evaporation following warning) *For simplicity, we have removed the statement indicating that the change in salinity could be associated with warming. This sentence is about trends at 50 m depth. Speculation on the source of salinity change should thus not be attributed to a process that would actually be taking place at the surface. While evaporation could be the source of increasing salinity, one*

*would have to discuss movement of the water mass at 50 m, and hypothesize that when this water mass was at the surface it was subject to evaporation. As we do not have knowledge on water mass movement, we have removed speculation on the change in salinity at 50 m.*

12. L486 … are still in agreement. (not: agreeable) *The sentence has been re-written as: "While the uncertainty for DYFAMED pH data is large, the trends are comparable to those observed at Point B".*

13. L488 I would suggest: This probably indicates that … *Done.*

14. L495 I suggest for clarity: … that may be occurring in more coastal regions of the Mediterranean. *Done.*

15. L496-499 I don't quite understand this. You write about a contrast between the western and eastern Med, but the acidification trends are very similar. *We have removed this sentence.*

16. L516 outgassing (instead of: off-gassing)? *Revised sentence reads "Summer warming leads to $pCO_2$ outgassing".*

17. L589 delete: a. *Done.*

18. Fig. 2g, 3g and 2n, 3n: the symbol on the y-axis is not correct. Also in other figures some symbols were not printed. Please check. *This occurred while converting the Word document to PDF. The high-resolution images do not have this issue.*

[revised manuscript text omitted]